# Spectral Adapter: Fine-Tuning in Spectral Space

**Fangzhao Zhang**
Electrical Engineering
Stanford University
zfzhao@stanford.edu

**Mert Pilanci**
Electrical Engineering
Stanford University
pilanci@stanford.edu

## Abstract

Recent developments in Parameter-Efficient Fine-Tuning (PEFT) methods for pretrained deep neural networks have captured widespread interest. In this work, we study the enhancement of current PEFT methods by incorporating the spectral information of pretrained weight matrices into the fine-tuning procedure. We investigate two spectral adaptation mechanisms, namely additive tuning and orthogonal rotation of the top singular vectors, both are done via first carrying out Singular Value Decomposition (SVD) of pretrained weights and then fine-tuning the top spectral space. We provide a theoretical analysis of spectral fine-tuning and show that our approach improves the rank capacity of low-rank adapters given a fixed trainable parameter budget. We show through extensive experiments that the proposed fine-tuning model enables better parameter efficiency and tuning performance as well as benefits multi-adapter fusion. Code is released at https://github.com/pilancilab/spectral_adapter.

## 1 Introduction

Size of language and vision model undergoes a drastic explosion in recent days and results in billions of parameters up to date. While fine-tuning has been used a lot for adapting pretrained large models to various downstream tasks, fine-tuning tasks become increasingly hard with current size of pretrained models due to the huge demand of computing resource. Meanwhile, exchange and storing of fine-tuned models are also expensive given their enormous size. To alleviate these rising problems for fine-tuning large pretrained models, a recent line of research has digged into the Parameter-Efficient Fine-Tuning (PEFT) model family and harnessed great attention. A high-level philosophy behind those PEFT methods is to train a reduced number of parameters compared to full fine-tuning, which instantly saves computing resource and enables light-weight fine-tuned model exchange. Among all PEFT methods, Low-Rank Adaptation (LoRA) [20] model is a huge success attributed to its simplicity and effectiveness. Specifically, LoRA proposes to tune an additive trainable low-rank matrix and brings zero inference latency after merging the adapter into pretrained model weights. Since its emergence, numerous variants of LoRA have been developed. For instance, AdaLoRA [65], IncreLoRA [62], and DyLoRA [54] propose to dynamically adjust LoRA rank distribution for improving tuning efficiency, QLoRA [10] combines LoRA with model quantization to further save computing resource, LoRA+ [16] and PrecLoRA [61] study the optimization landscape of LoRA training, and more recent variant DoRA [32] decomposes pretrained weights into magnitude and direction components and applies LoRA for direction tuning, see Apppendix A for a more comprehensive review of different LoRA variants. Other PEFT methods such as Orthogonal Fine-Tuning (OFT) proposes to multiply pretrained weights by tunable orthogonal matrices for preservation of hypersphere energy between pretrained neurons. Though these different PEFT methods focus on improving fine-tuning efficiency with reduced parameters, rare attention has been paid to utilize pretrained model weights' information beyond its magnitude in the fine-tuning procedure.

38th Conference on Neural Information Processing Systems (NeurIPS 2024).

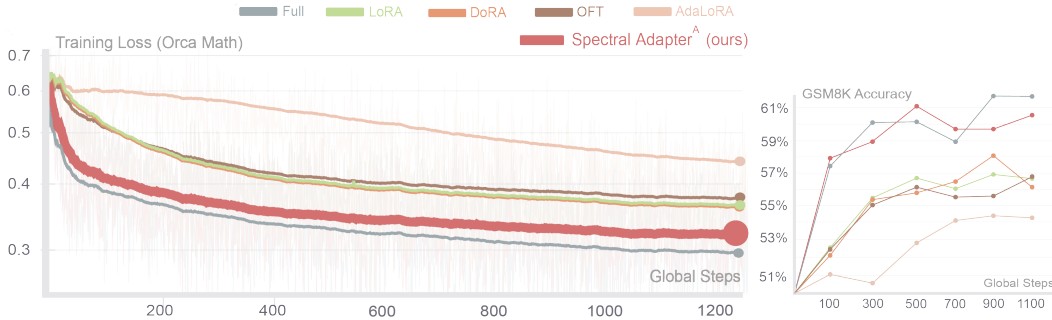

Figure 1: Training loss of fine-tuning Llama3 8B model with Orca Math dataset [38] and evaluation score on GSM8K benchmark [8]. We follow experimental setup in [53], see Appendix F.1 for details. All methods except full fine-tuning maintain approximately $0.23\%$ trainable parameters.

Prior research in statistical machine learning such as [36] has studied the Empirical Spectral Distribution (ESD) of deep models' weight matrices and found that the ESDs for larger model weights are usually more structured and contain indicative information to distinguish between different training stages. More recent work such as [4] investigates the "dark matter" effect of bottom spectral space of model weights and recognizes its critical role in attention sink phenomenon observed in [57]. Both work contributes to decrypting spectral information of model weights and sheds light on building insightful understanding of the connection between weight matrices' spectral information and model performance. In this work, we explore further the value of model weights' spectral pattern and unravel its effectiveness in enhancing fine-tuning tasks. We showcase via extensive empirical observation that integration of spectral information of pretrained model weights improves current PEFT methods' parameter efficiency, tuning effect, and arises as a natural solution to multi-adapter fusion problems. Moreover, the suggested fine-tuning model maintains better practicality compared to prior spectral tuning models, which will be investigated further below.

Though any technique for weight fine-tuning can be directly applied to fine-tune singular vector matrices of pretrained model weights, we investigate two specific forms of such extension, namely additive tuning and orthogonal rotating the top singular vector space, which we address as Spectral Adapter$^A$ and Spectral Adapter$^R$ respectively in later content. The spectral adaptation mechanisms being considered are formally depicted in Section 2. As a warmup, to show that incorporating spectral information is indeed helpful, Figure 1 displays the training loss of fine-tuning Llama3 8B model on HuggingFace Orca Math dataset and validation score on GSM8K benchmark, from which it can be clearly observed that Spectral Adapter$^A$ performs superior to recent variants of PEFT methods and behaves closest to full fine-tuning, here we follow experimental setup in [53], see Appendix F.1 for details and more investigation. In below, we first introduce the fine-tuning model being studied in Section 2 and we then provide some theoretic insights in Section 3. After that, we detail the advantage of our spectral adapter in enhancing fine-tuning result, improving model's parameter efficiency, and helping with multi-adapter fusion as well as address any concern with respect to practicality issues in Section 4. Conclusion and future work is discussed in Section 5. For sake of page limitation, literature review is deferred to Appendix A.

To summarize, the proposed spectral adaptation mechanism demonstrates the first attempt to fine-tune spectral space of pretrained model weights in a parameter-efficient and storage-economic way which improves current PEFT methods from aspects involving tuning results, parameter efficiency, and multi-adapter fusion. We hope this work serves as a building block and motivates further and deeper insightful investigation for exploring spectral structure of pretrained model weights, which becomes increasingly meaningful especially in current large model regime.

## 2 Spectral Adapter: Incorporating Spectral Information into Fine-Tuning

Motivated by the intrinsic low-rank of weight shifts in fine-tuning procedure studied in [2], LoRA [20] proposes to add a low-rank factorized trainable matrix to pretrained model weights and tune only these additive parameters for downstream task adaptation, which usually injects far fewer trainable

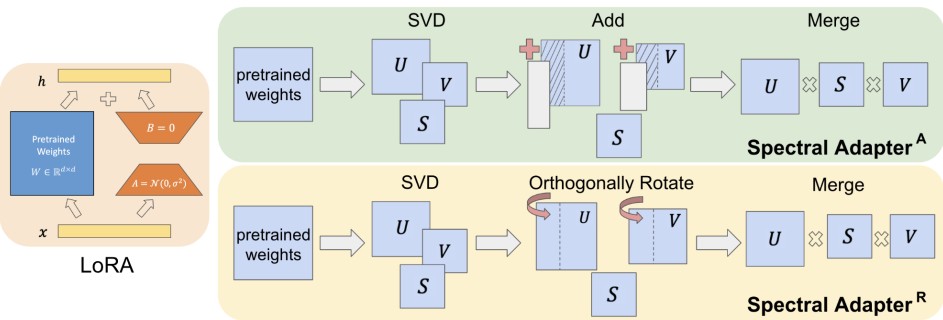

Figure 2: Compared to LoRA which proposes to add low-rank trainable matrices to pretrained weights, we study two types of spectral adapters: **Spectral Adapter**$^A$ considers additively tuning the top columns of singular vector matrices and **Spectral Adapter**$^R$ considers orthogonally rotating the top columns of singular vector matrices.

parameters compared to full fine-tuning and results in light-weight tuned adapters. LoRA serves as an outstanding representative of PEFT family and is now widely-used for different fine-tuning tasks. Inspired by the parameter efficiency of LoRA and the close connection between matrix rank and its spectral representation, here we study two spectral fine-tuning mechanisms, both are completed via first carrying out Singular Value Decomposition (SVD) of pretrained model weights and then fine-tuning the top columns of singular vector matrices obtained via the SVD. More precisely, consider a pretrained weight matrix with its spectral representation of form $W = USV^T$, we define additive spectral adapter as

$$\textbf{Spectral Adapter}^A(W) \coloneqq [U_1 + A_U \ U_2]S[V_1 + A_V \ V_2],$$

and correspondingly the rotational version

$$\textbf{Spectral Adapter}^R(W) \coloneqq [U_1 R_U \ U_2]S[V_1 R_V \ V_2],$$

where $U_1, V_1$ denote the top-$r$ columns of $U$ and $V$ and $U_2, V_2$ denote the rest of the columns. $A = (A_U, A_V)$ consists of trainable matrices of shape same as $(U_1, V_1)$ and $R = (R_U, R_V)$ consists of two trainable orthogonal matrices of shape $r$ by $r$ such that $R_U^T R_U = R_V^T R_V = I$. As we show in later sections, the orthogonality constraint is efficiently handled with the Cayley parameterization, see Section 4.3 for details. The proposed fine-tuning model architecture can be visualized from Figure 2. Here Spectral Adapter$^A$ more resembles LoRA as it is of additive form while Spectral Adapter$^R$ more resembles prior Orthogonal Fine-Tuning (OFT) method which we compare further in Section 4. To ensure zero initialization as often done for PEFT methods, we initialize $A_U$ and $A_V$ both at zero. For rotational spectral adapter, we initialize $R_U$ and $R_V$ as identity matrices.

A more thorough literature review suggests that prior work considering tuning model weights' spectral representation (FSGAN[47], SVDiff [15]) has been proposed for alleviating overfitting when fine-tuning different vision models. These methods only look at tuning the singular values of flattened CNN weights and thus have fixed amount of trainable parameters. Moreover, these methods require storing all $U, S$ and $V$ during training while only the diagonal vector of $S$ is tuned, which nearly doubles the storage requirement compared to pretraining when fine-tuning on downstream tasks. Contrarily, we consider incorporating spectral information in generic fine-tuning procedure for different layers (flattened CNN weights, dense linear weights, etc.) and our method enables flexible parameter budget choices by varying values of $r$. Methodology-wise, we consider tuning the top-$r$ columns of $U$ and $V$ by additive and rotational tuning, both requiring only these top columns to be stored additionally and the left part can be merged into a single weight matrix. See Section 4.4 for more investigation on practicality of the proposed method.

## 3  Theoretical Insights

After introducing the model architecture of spectral adapter we consider, the main question now remains whether tuning the spectral representation of pretrained weights is indeed an improvement over existing PEFT methods. Before we step into our empirical observations, we first provide

some theoretical insights for the proposed spectral adaptation mechanism. In this section, we show advantage of our spectral adapter method compared to LoRA from two theoretic perspectives by analyzing both the rank capacity of the adapters (Section 3.1) and the subspace alignment of pretrained weight matrices (Section 3.2). Specifically, we will see that Spectral Adapter$^A$ has larger rank capacity than LoRA adapter, which indicates the tuned weight has more adaptation freedom and thus is more desirable. Moreover, the dominant spectral direction of pretrained weight matrix identifies more ideal neuron alignment under the setting we consider in Section 3.2, which justifies the robustness of tuning top singular vectors in our spectral adapter. In Appendix D, we show that Spectral Adapter$^A$ is approximately equivalent to DoRA [32] for vector-form weights.

## 3.1 Adapter Rank Capacity

For any pretrained weight matrix $W$, suppose that the adapter is given by the parameterization $f_\theta(W)$ where $\theta$ represents trainable weights. For instance with LoRA adapter, $f_\theta(W) = W + AB^T$, where $\theta = \{A, B\}$ is trainable. We define the *rank capacity* of an adapter $f_\theta(W)$ as follows:

$$\mathcal{R}(f_\theta; W) := \max_\theta \mathbf{rank}(f_\theta(W)) - \min_\theta \mathbf{rank}(f_\theta(W)),$$

which describes the range of matrix ranks the tuned weight can achieve given a specific adapter form. Then, the following lemma shows that Spectral Adapter$^A$ has twice the rank capacity of LoRA adapter under an equal number of trainable parameters.

**Lemma 3.1.** *Suppose that $W \in \mathbb{R}^{n \times m}$ is an arbitrary full row-rank matrix and $n \le m$ without loss of generality. Consider rank-r LoRA and rank-r additive spectral adapter, which have an equal number of trainable parameters. We have*

$$\mathcal{R}(\mathrm{LoRA}; W) = r,$$

$$\mathcal{R}(\mathrm{Spectral\ Adapter}^A; W) = 2r.$$

See Appendix B for proof. Therefore when pretrained model weight matrix is close to full row-rank, as what has been observed in [20], Spectral Adapter$^A$ has nearly double rank capacity compared to LoRA adapter. Furthermore, some prior work explicitly imposes low-rank constraint when training original NNs [50, 43, 66, 22, 68, 24, 9]. Using LoRA adapter to fine-tune such pretrained model weights would destroy their rank constraints while applying spectral adapter preserves the constraints.

Next we proceed to show that top spectral space of pretrained weight matrices is more aligned with ideal neuron direction under a simple setting via subspace decomposition analysis of pretrained model weights. This observation corroborates our choice of tuning top singular vectors in our proposed spectral adaptation mechanism. Empirically, we observe that tuning top directions performs superior to tuning bottom ones, see Appendix F.3 and F.5.1 for related experiments.

## 3.2 Weight Subspace Alignment

Consider two-layer ReLU network with $m$ hidden nodes and univariate output. For squared loss objective, we can write out the training problem explicitly as

$$\min_{W^{(1)}, W^{(2)}} \|(XW^{(1)})_+ W^{(2)} - y\|_2^2 + \beta(\|W^{(1)}\|_F^2 + \|W^{(2)}\|_2^2),$$

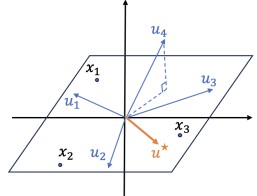

where $X \in \mathbb{R}^{n \times d}$ is the data matrix, $(W^{(1)} \in \mathbb{R}^{d \times m}, W^{(2)} \in \mathbb{R}^m)$ are first and second layer weights respectively and $y \in \mathbb{R}^n$ is the label vector. For better visualization, we take $d = 3$. Consider the case that all data points lie on $xy-$plane, which mimics the usual observation that data points occupy a low-dimensional manifold. Then we can decompose each first layer neuron $W_j^{(1)} \in \mathbb{R}^d$ into $W_j^{(1)} = w_{j1} + w_{j2}$ where $w_{j1} \in \mathcal{R}(X), w_{j2} \perp \mathcal{R}(X)$. With simple algebra, for non-zero weight decay which is often the default setting for current deep learning optimizers, one can derive $w_{j2} = 0$ and thus $W_j^{(1)} = w_{j1} \in \mathcal{R}(X)$. Therefore all optimal neurons lie also in $xy-$plane. However, due to optimization errors, some

Figure 3: Top singular vector of pretrained weight recognizes more ideal neuron direction. Illustration plot for Section 3.2.

of the trained neurons might be slightly deviated from $xy-$plane, as illustrated in Figure 3, where $u_i$ indicates pretrained neuron directions, though most of them lie in $xy-$plane, some might deviate (i.e., $u_4$). $u^\star$ indicates the top singular vector direction of pretrained weight $W^{(1)}$ which here recognizes the $xy-$plane orientation, and thus fine-tuning $u^\star$ is noiseless and is expected to be more robust.

# 4 Empirical Results: The Impact of Spectral Information

We experiment our proposed spectral adapter with fine-tuning large language models and diffusion models and compare against various recent PEFT methods. From language model experiments, we observe that Spectral Adapter$^A$ performs superior to various PEFT baselines and harnesses higher scores on different benchmarks, which again verifies the effectiveness of incorporating spectral information into the fine-tuning procedure, see Section 4.1 for details. For diffusion model experiments, we will see that the advantage of spectral adapter comes in two-fold: Spectral Adapter$^A$ offers a natural solution to existing problems in multi-adapter fusion procedure and Spectral Adapter$^R$ manifests finer-grained parameter budgets as well as better parameter efficiency, see Section 4.2 and 4.3 respectively. For a fair comparison with all baselines, we use their official implementation and follow hyperparameter setting in their original reports as long as available. See each individual section for corresponding experimental details. All experiments are done with NVIDIA RTX A6000 GPU.

## 4.1 Language Model Fine-Tuning: Enhancing Fine-Tuning Results with Spectral Adapter$^A$

For large language model experiments, we present experimental results for fine-tuning DeBERTaV3-base model (185M) and Mistral model (7B) on GLUE and GSM8K tasks respectively. Our Spectral Adapter$^A$ method achieves superior tuning results compared to various recent PEFT methods in most experiments.

**DeBERTaV3-base Experiment.** Table 1 shows fine-tuning results of DeBERTaV3-base model on GLUE benchmarks with various PEFT methods. For a fair comparison, we use official implementations for LoRA, DoRA, OFT and AdaLoRA in HuggingFace PEFT library, with hyperparameter setting for LoRA [20] and AdaLoRA [65] following their original reports. We use same hyperparameter setting as LoRA for DoRA and follow the setting used in BOFT [33], a variant of OFT, for OFT experiments. We abbreviate Spectral Adapter$^A$ as Spectral$^A$ for presentation simplicity and we tune hyperparameters for Spectral Adapter$^A$. See Appendix F.2 for hyperparameter details and F.3 for loss/validation plot comparison. We fine-tune all $q, k, v$ matrices in attention layers. Our Spectral Adapter$^A$ achieves highest average score and best scores for most tasks with fewest trainable parameters.

| Method | # Param | GLUE | | | | | | | | |
| --- | --- | --- | --- | --- | --- | --- | --- | --- | --- | --- |
| | | MNLI | SST-2 | MRPC | CoLA | QNLI | QQP | RTE | STS-B | Avg. |
| LoRA$_{r=24}$ | 0.72% | 88.87 | 95.06 | 87.00 | 65.84 | 91.87 | 91.45 | 81.22 | 90.43 | 86.47 |
| DoRA$_{r=24}$ | 0.73% | 88.91 | 95.29 | 88.72 | 65.84 | 92.01 | 91.51 | 80.14 | 90.10 | 86.57 |
| OFT$_{r=4}$ | 0.72% | 89.16 | 95.06 | 87.74 | 66.75 | 93.28 | 91.33 | 78.70 | 89.72 | 86.47 |
| AdaLoRA$_{r=24}$ | 1.07% | 89.44 | 94.95 | 89.70 | 63.06 | 93.17 | 91.48 | **83.75** | **91.22** | 87.10 |
| Spectral$^A_{r=24}$ | 0.72% | **89.79** | **95.75** | **90.19** | **69.44** | **93.35** | **91.65** | 83.39 | 90.64 | **88.03** |

Table 1: Accuracy comparison of fine-tuning DeBERTaV3-base with various PEFT methods on GLUE benchmarks. Spectral$^A$ is abbreviation for Spectral Adapter$^A$. See Section 4.1 for experimental details.

**Mistral 7B Experiment.** We experiment our Spectral Adapter$^A$ with Mistral 7B model [23] fine-tuned for GSM8K task [8]. Since all baseline model reports include no fine-tuning tasks with the Mistral family, we use official implementations of all baseline methods for comparison and we fix learning rate to be $2.5e - 5$ for all methods following [51].

We take $r = 8$ for LoRA, DoRA and Spectral Adapter$^A$ to maintain approximately same number of trainable parameters for all methods. Table 2 presents the accuracy comparison where Spectral$^A$ stands for Spectral Adapter$^A$. From the result, we observe that our Spectral Adapter$^A$ scores higher than both LoRA and DoRA by a large margin and increases the pretrained model baseline significantly, which verifies the effectiveness of the proposed spectral adaptation mechanism. See Appendix F.4 for more about experi-

| Method | #Param | GSM8K |
| --- | --- | --- |
| Pre-Trained | – | $37.91 \pm 1.34$ |
| LoRA$_{r=8}$ | 0.16% | $44.81 \pm 1.37$ |
| DoRA$_{r=8}$ | 0.17% | $43.82 \pm 1.37$ |
| Spectral$^A_{r=8}$ | 0.16% | $49.73 \pm 1.38$ |

Table 2: Accuracy comparison of fine-tuning Mistral 7B model with different PEFT methods on GSM8K benchmark. See Section 4.1 for experimental details.

mental details. Note for a different learning rate, DoRA performs better than LoRA while still worse than our method, see also Appendix F.4 for details.

## 4.2 Diffusion Model Fusion: Improving Multi-Object Fine-Tuning with Spectral Adapter[A]

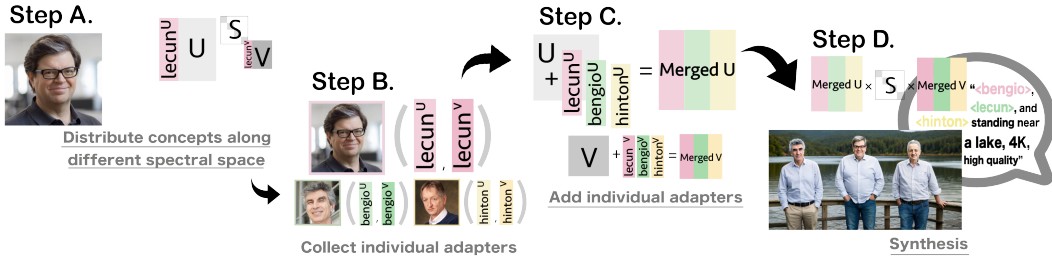

Figure 4: Distributing different concept tunings along different spectral space helps with identity preservation in multi-adapter fusion, see Section 4.2 for details.

Multi-adapter fusion is a current bottleneck in diffusion model fine-tuning tasks with LoRA adapters. Simply adding different LoRA adapters tuned for distinct objects will result in problems involving identity loss and concept binding [12]. To tackle this toughness, different methods emerge such as Gradient Fusion [12] and Orthogonal Adaptation [42]. Specifically, Orthogonal Adaptation method proposes to fix LoRA parameter $B$ to have orthogonal basis and train $A$ solely. Experiments there show that merging LoRA weights with such orthogonal basis helps preserving individual object characteristics compared to its non-orthogonal counterpart. In Orthogonal Adaptation [42], the authors maintain $B$ by manually keeping large orthogonal matrices for different layer sizes and sample $r$ columns from corresponding orthogonal matrix to form $B$ for each LoRA adapter. With knowledge from random matrix theory, such sampled matrices are likely to have orthogonal basis.

Notably, our Spectral Adapter[A] naturally operates on orthogonal singular vectors and thus introduces an elegant solution to multi-adapter fusion problems by distributing different concept tunings along different columns of singular vector matrices, which maps to wireless communications where the signals are distributed over non-overlapping frequencies. A subtlety here lies in the choice of column space for different fine-tuning tasks: (1) Sample-based methods can be adopted if data privacy is considered and different tuning tasks are done independently. In Appendix F.5, we show that tuning top columns manifests better generation quality compared to both tuning bottom columns and sampling random orthogonal basis as what has been done in Orthogonal Adaptation [42]. Thus there is a trade-off between high-quality generation and concept collapsing, i.e., sampling from top singular vectors is more encouraged while column overlapping between concepts happens more often compared to sampling from the whole set. (2) On the other hand, if fine-tuning tasks are not isolated and can collaborate on the column scheduling, then more deliberate tuning scheduling can be adopted, for example in a two-concept tuning task with $r = 4$, the first concept can allocate first to fourth columns and the second concept then claims fifth to eighth columns. Figure 4 demonstrates steps for the same method for three-concept tuning task. Since we expect fine-tuned weights to stay close to original weights, though both row space and column space are tuned in spectral adapter, this adaptation mechanism approximates orthogonal-basis tuning for different objects and thus we expect it helps improving identity preservation for multi-adapter fusion. In this section, we investigate this effect via extensive diffusion model experiments.

Our experiments follow [42] and build on [12] which studies multi-LoRA fusion. We experiment with multi-object tuning and face generation tasks. Due to space limitation, we present some multi-object tuning results below and we leave the rest to Appendix F.5. For all tasks, we compare against baselines including Gradient Fusion [12], Orthogonal Adaptation [42], and FedAvg [37]. We start with a simple review for these baseline methods.

**Baseline Review**

To merge different LoRA adapters, say we have a set of LoRA parameters $\{\Delta\theta_1, \ldots, \Delta\theta_n\}$ where $\Delta\theta_i = A_i B_i^T$ and pretrained parameter $\theta_0$, FedAvg [37] proposes to merge them in to a single parameter by taking a weighted average as $\theta_{\text{merged}} = \theta_0 + \sum_i \lambda_i \Delta\theta_i$, where $\lambda_i$ is the weight attached to parameter $\Delta\theta_i$ and is usually taken to satisfy $\sum_i \lambda_i = 1$, i.e., $\theta_{\text{merged}}$ is a convex combination of individual adapters. Gradient Fusion [12] instead considers solving an auxiliary optimization problem of form $\theta_{\text{merged}} = \text{argmin}_\theta \sum_{i=1}^n \|(\theta_0 + \Delta\theta_i)X_i - \theta X_i\|_F^2$ where $X_i$ represents the input activation of the $i$-th concept. Orthogonal Adaptation [42] follows FedAvg method and replaces original LoRA

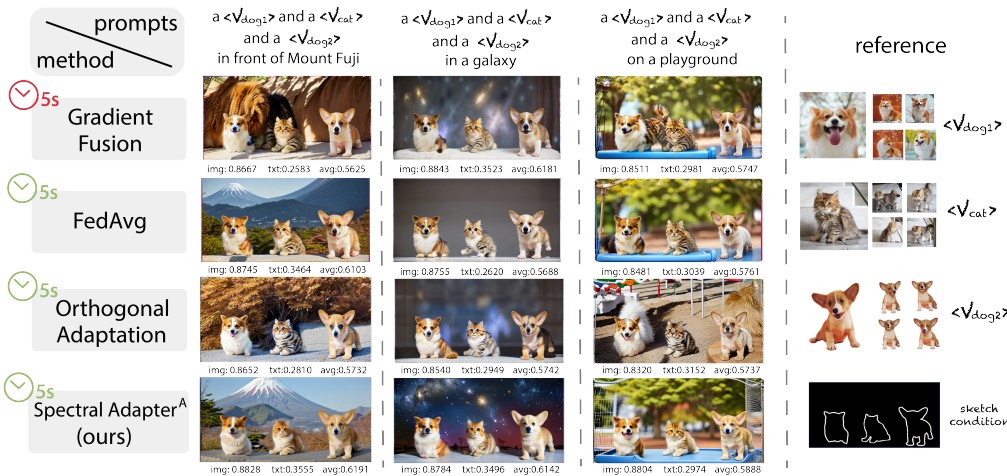

Figure 5: Generation results of Chilloutmix diffusion model [1] with different fused adapters tuned on three custom animal concepts. See Section 4.2 for details.

parameters with orthogonal-based LoRA adapters. For our method, to merge different spectral adapters, let $\theta_0 = U_0 S_0 V_0^T$ denote the spectral representation of pretrained model weight. Given a set of spectral adapters $\{(U_i, V_i), \ldots, (U_n, V_n)\}$ with zero-padding to make the shape the same as $(U_0, V_0)$, we follow FedAvg and compute $\theta_{\text{merged}} = (U_0 + \sum_i \lambda_i U_i) S_0 (V_0 + \sum_i \lambda_i V_i)^T$. In the following experiments, we take $\lambda_i = 1/n$ as in [42] for all FedAvg, Orthogonal Adaptation, and our Spectral Adapter$^A$ fusion. Notably, all FedAvg, Orthogonal Adaptation, and our Spectral Adapter$^A$ fusion can be done approximately instantly while Gradient Fusion usually takes around $10 \sim 15$ minutes for solving its auxiliary optimization problems for all concept adapters.

**Multi-Object Generation**

We follow default training setting in [12] and fine-tune the Chilloutmix diffusion model [1] on three custom animal concepts, see original animals in "reference" in Figure 5. For better spatial alignment, we adopt T2I-Adapter [39] with sketch condition and we set guidance equal to one, see also "reference" in Figure 5 for the sketch condition being used. LoRA rank $r = 8$ is adopted. For baseline comparisons, we use original code for Gradient Fusion [12] and Orthogonal Adaptation [42]. We adapt code of Gradient Fusion for FedAvg method since there is no official implementation available. Custom animal name is replaced with special token $<V_{\text{animal}}>$ for fine-tuning. For our Spectral Adapter$^A$, we follow the method depicted in Figure 4 and tune first, second, and third top eighth columns of singular vector matrices for different animal concepts. Figure 5 shows the generation results with different methods for selected prompts. Notably, baseline methods sometimes fail to capture the custom animal concepts while Spectral Adapter$^A$ recognizes all custom animals and generates visually satisfactory images. For better measurement, we also compute the alignment scores for each generated image with both reference images and prompt texts. It can be witnessed that our method achieves better alignment scores compared to baselines. See Appendix F.7 for details on alignment score computation.

### 4.3 Diffusion Model Expressiveness: Improving Parameter Efficiency with Spectral Adapter$^R$

Spectral Adapter$^R$ is closely connected to prior Orthogonal Fine-Tuning (OFT) [45] method which proposes to multiply the pretrained model weights by trainable orthogonal matrices in the fine-tuning procedure. Motivation behind OFT is to preserve hyperspherical energy which characterizes the pairwise neuron relationship on the unit hypersphere. Unlike OFT which orthogonally rotates neurons, Spectral Adapter$^R$ multiplies the top-$r$ columns of singular vector space $U$ and $V$ by orthogonal trainable matrices. For our implementation, several options are available for maintaining a trainable orthogonal matrix such as adding an orthogonality penalty in the objective function considered in [65] or via Cayley parameterization considered in [45]. We follow [45] and adopt Cayley parameterization which is supported by Pytorch [44]. Specifically, the orthogonal matrix $R$ is constructed via $R = (I + Q)(I - Q)^{-1}$ with a skew-symmetric matrix $Q$ maintained as $(A - A^T)/2$

where $A$ is our trainable parameter. Compared to adding an auxiliary orthogonality penalty, this parametrization is exact and thus the SVD form is preserved after tuning with Spectral Adapter$^R$ and can be adopted directly for subsequent fine-tuning tasks, which we state formally as a lemma below:

**Lemma 4.1.** *With the Cayley parametrization, Spectral Adapter$^R$ is an exact rotation operation and thus preserves the structure of the SVD of the fine-tuned weight. Subsequent fine-tunings can be applied consequently without recomputing the SVD each time.*

See Appendix C for the proof of above lemma. Unlike LoRA which requires number of trainable parameters to scale with weight size, when tuning top-$r$ columns of $U$ an $V$, Spectral Adapter$^R$ only requires two trainable matrices of size $r \times r$ and thus can be more parameter-efficient especially for large pretrained weight. For common weight size such as $W \in \mathbb{R}^{1024 \times 1024}$, LoRA with only $r = 1$ introduces same number of trainable parameters as Spectral Adapter$^R$ with $r = 32$. For a thorough analysis on parameter efficiency improvement brought by Spectral Adapter$^R$, we here also compare with different variants of LoRA which are proposed for trainable parameter savings. We review all baselines in detail below.

**Baseline Review**

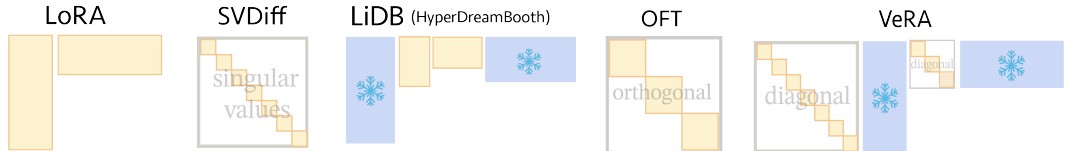

We compare our Spectral Adapter$^R$ with LoRA [20], SVDiff [15], LiDB [48], OFT [45], and VeRA [25]. Though the other methods are proposed for vision model tuning, VeRA is originally proposed for LLM tuning and we extend it here to diffusion model tuning due to its parameter efficiency. Consider a pretrained weight $W \in \mathbb{R}^{n \times n}$, SVDiff originally proposes to tune all singular values of flattened CNN weights, here we extend it to tune all singular values of text encoder and U-Net weights for our comparison, thus trainable parameter attached to $W$ will be of size $n$ and is nonadjustable. LiDB stands for Lightweight Dreambooth and proposes to cut down trainable parameter budget by introducing auxiliary frozen matrix $A_{\text{aux}} \in \mathbb{R}^{n \times a}$ and $B_{\text{aux}} \in \mathbb{R}^{b \times n}$, then it mimics LoRA but uses $A_{\text{aux}} AB^T B_{\text{aux}}$ in replace of $AB^T$ with trainable ($A \in \mathbb{R}^{a \times r}, B \in \mathbb{R}^{b \times r}$). Thus with $a, b < n$, LiDB requires $(a + b)r < 2nr$ trainable parameters. In below, we use $a = 50, b = 100$ as default in [48]. OFT multiplies the weight matrix by a trainable orthogonal matrix via Cayley parametrization discussed above, thus its complete version requires $n^2$ trainable parameters. For parameter efficiency, OFT proposes to use block-diagonal trainable matrix with all diagonal blocks being orthogonal. Thus with $r$ diagonal blocks, the number of trainable parameter will be $r \times (n/r)^2$.

Further reduction of trainable parameter is achieved via sharing the diagonal blocks, which demands only $(n/r)^2$ parameters. In below comparison, we use this shared block-diagonal version for best parameter efficiency of OFT. VeRA proposes to use $\Lambda_a A \Lambda_b B^T$ in replace of $AB^T$ where $\Lambda_a$ and $\Lambda_b$ are diagonal matrices of size $n \times n$ and $r \times r$ respectively. Thus the total number of trainable parameters by VeRA is $(n + r) \propto n$. Table 3 compares dif-

| Method | | Granularity | #Param | Auxiliary Param |
|---|---|---|---|---|
| LoRA | ☹ | $\infty$ | $2nr \propto n$ | no |
| SVDiff | ☹ | 1 | $n \propto n$ | no |
| LiDB | ☹ | $\infty$ | $(a + b)r \propto r$ | yes |
| OFT | ☹ | # factors of $n$ [1] | $(n/r)^2 \propto \frac{n}{r}$ | no |
| VeRA | ☹ | $\infty$ | $n + r \propto n$ | yes |
| Spectral Adapter$^R$ | ☺ | $n$ | $2r^2 \propto r$ | no |

[1] Ceiling operation is ignored for this count.

Table 3: Baseline methods comparison for parameter efficiency. Granularity indicates number of trainable parameter budgets available. See Section 4.3 for details.

ferent properties across all methods, where $n$ represents weight size and $r$ represents rank for all methods except for OFT, where $r$ denotes number of diagonal blocks.

**Parameter Efficiency**

We fine-tune the Chilloumix diffusion model [1] with various PEFT methods on custom vase concept and present the generation results for prompt "a <$V_{\text{vase}}$>" in Figure 6 for various trainable parameter budgets, where grey dash denotes that the corresponding parameter budget is unobtainable with a given adapter no matter how the hyperparameter is chosen and empty entry without grey dash

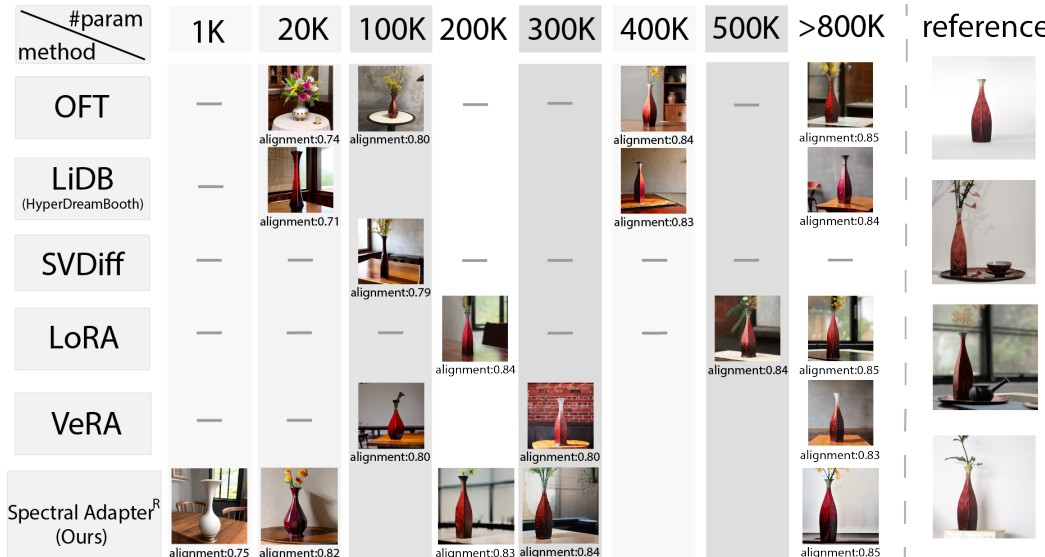

Figure 6: Generation results for prompt "a <$V_{vase}$> on a table" after fine-tuning Chilloutmix diffusion model [1] on custom vase images with different PEFT methods. See Section 4.3 for details.

represents that there is a way to achieve the corresponding parameter budget though the generation result is skipped for better visualization. We follow default LoRA implementation in [12] for LoRA baseline and adjust it for all other methods. From Figure 6, it can be observed that LoRA, OFT, and LiDB start to generate vase close to custom vase with at least $200k$ trainable parameters. SVDiff and VeRA are unable to generate ideal vase images even if scaled to large parameter budget. On the contrary, Spectral Adapter$^R$ starts to recognize the custom vase concept with only $20k$ trainable parameters and has finer-grained parameter choices compared to other methods, i.e., notably Spectral Adapter$^R$ can have as few as $1k$ parameters while other methods start with at least tens of thousands of trainable parameters. In a word, Spectral Adapter$^R$ enjoys finer-grained parameter budget choices and manifests better visual quality with fewer parameters, thus achieves enhanced parameter efficiency compared to various other PEFT methods.

Figure 7 below presents generation results of Chilloutmix diffusion model [1] tuned on custom chair concept with different methods under various parameter budgets. The prompt used is "a yellow <$V_{chair}$>". See "reference" in Figure 7 for original chair images. From the generation results, it can be observed that LoRA generates reasonable chairs for all rank $r = 1, 2, 3$ though it already induces $273k$ parameters even if rank is set to 1. OFT and VeRA start to recognize custom chair with $> 100k$ parameters. SVDiff has a single fixed trainable parameter budget of size around $100k$. LiDB forms a

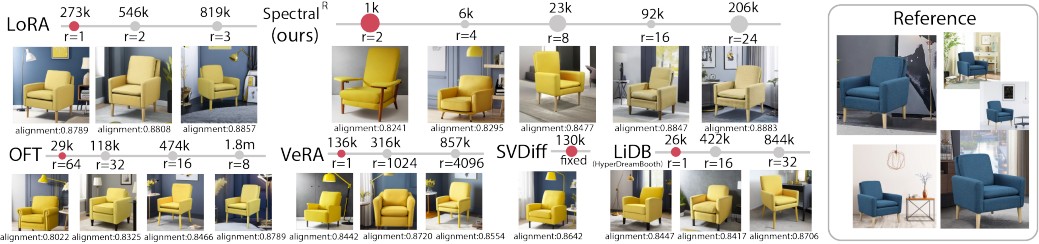

Figure 7: Generation results for prompt "a yellow <$V_{chair}$>" after fine-tuning Chilloutmix diffusion model [1] on custom chair images with different PEFT methods. Spectral$^R$ is abbreviation for Spectral Adapter$^R$. See Section 4.3 for details.

competitive candidate and generates satisfactory images with smallest trainable parameter budget among all baseline methods. However, our Spectral Adapter$^R$ still generates images better aligned to

reference images with as few as $20k$ trainable parameters and has finer-grained parameter budget choices compared to LiDB. See Appendix F.6 for hyperparameter setting and Appendix F.7 for alignment score computation details.

## 4.4 Final Note: A Closer Look at SVD Cost

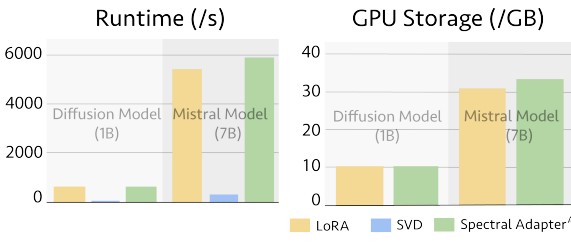

Figure 8: Runtime and GPU storage cost plot. See Section 4.4 for details.

To alleviate the concerns with respect to online training cost and show that our proposed method is very practical, we provide runtime and GPU storage cost bar plot in Figure 8, which shows runtime and GPU storage cost for LoRA and for our Spectral Adapter[A] when used for fine-tuning diffusion model in Section 4.2 and Mistral 7B model in Section 4.1. Here we adopt rank $r = 8$ for both LoRA and Spectral Adapter[A]. It can be observed that our Spectral Adapter[A] introduces negligible runtime and storage overhead for current large model size. Modern numerical tools such as randomized SVD [13] can also be exploited for further runtime reduction and the SVD procedure can be parallelized when multiple machines are available. See Appendix E for further investigation.

## 5 Conclusion and Limitations

In this work, we investigate the incorporation of spectral information of pretrained model weights into current PEFT models by introducing a spectral adaptation mechanism which updates only the top singular vectors of pretrained weights. We investigate the additive and rotational variants of such spectral adaptation mechanism. Theoretically, we show the motivation of tuning top singular vectors by comparing the rank capacity of different fine-tuning models and carrying out weight decomposition of pretrained model layers. Empirically, we verify the superiority of our proposed spectral adaptation method compared to various recent PEFT methods from different aspects via extensive experiments. To our best knowledge, this is the first work considering incorporating spectral information as a practical generic paradigm for fine-tuning tasks and enhances fine-tuning results, parameter efficiency, as well as benefits multi-adapter fusion of existing PEFT methods. For future work, fine-tuning spectral representation of different components, i.e., only the attention layer, of current large models is also worth studying. Other PEFT methods such as AdaLoRA [65] can also be dynamically combined with spectral adaptation.

A limitation of the current work remains in the choice of tuning top spectral space. Though its validity has been theoretically verified under simple settings, further investigation on tuning different columns of singular vector matrices is critical to understanding the role of spectral information in fine-tuning procedure. Besides, fine-tuning spectral representation of different components, i.e., only the attention layer, of current large models is also worth studying. Moreover, the time consumption of singular value decomposition procedure increases as model grows larger and thus faster singular value decomposition method also benefits.

# 6 Acknowledgements

This work was supported in part by the National Science Foundation (NSF) under Grant DMS-2134248; in part by the NSF CAREER Award under Grant CCF-2236829; in part by the U.S. Army Research Office Early Career Award under Grant W911NF-21-1-0242; and in part by the Office of Naval Research under Grant N00014-24-1-2164.

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

# Appendix

## A  Prior Work

Here we provide an overview of recent PEFT methods. Dating back to 2019, Houlsby et al. [19] develop the idea of parameter-efficient fine-tuning and introduce Adapter model, which injects trainable components between pretrained model layers, though the number of trainable parameters has been reduced due to the small size of adapters, this method incurs inference latency and is thus not desirable. Later improvement of Adapter fine-tuning focuses on improving inference latency [49, 26], fusing multiple adapters [7, 41, 18], modifying adapter model architecture [67], introducing parallelism [17, 69], and creating task-specific and layer-specific adapter [35, 30]. Another line of fine-tuning is prompt-tuning [27] which usually adds the trainable components into the prompt. Variants of prompt-tuning involve WARP [14], prefix-tuning [28], P-tuning [34], and ATTEMPT [3] which consider injecting different forms of trainable components. Multitask prompt-tuning is considered in [55, 56].

The more relevant PEFT methods to our spectral adaptation mechanism involves LoRA [20] and OFT [45], which inspires our Spectral Adapter$^A$ and Spectral Adapter$^R$ respectively. LoRA originates from the observation that model fine-tuning is intrinsically low-rank [2]. Variants of LoRA involve different methods proposing dynamic allocation of LoRA rank budgets [54, 62, 65, 6]. LoRA has been combined with model pruning [64] and quantization [10, 59, 29]. Some other variants further cut down the trainable parameter budget or activation storage by modifying LoRA model [25, 11, 63]. DoRA [32] fixes LoRA's low-rank limitation by decomposing pretrained model weights and isolating their magnitudes. Laplace-LoRA [60] incorporates Bayesian inference into LoRA parameters to improve calibration. LoRAHub [21], MOELoRA [31], and L-LoRA [52] consider multitask LoRA. Delta-LoRA [70] updates pretrained weights simultaneously from information of LoRA parameters. GLoRA [5] generalizes LoRA by introducing a prompt module. Another line of variants focuses on analyzing the optimization scheme of LoRA model [61, 16]. OFT studies the multiplicative fine-tuning and its variant BOFT [33] improves OFT by utilizing butterfly parametrization for better information delivery efficiency. [58] offers a comprehensive review of recent development of PEFT methods.

## B  Rank Capacity Proof

*Proof.* Consider weight matrix $W \in \mathbb{R}^{n \times m}$ with $n \leq m$ of full row rank. For LoRA parameter $A \in \mathbb{R}^{m \times r}, B \in \mathbb{R}^{n \times r}$ with $n \geq r$, final weight matrix $W + AB^T$ has rank in $[n - r, n]$. With Spectral Adapter$^A$ parameters $A_S \in \mathbb{R}^{m \times r}, B_S \in \mathbb{R}^{n \times r}$ where $n \geq 2r$. Let $X_r$ denote the first $r$ columns of any matrix $X$ and $X_{-r}$ denote the rest columns, final weight matrix $((U_r + A_S)S_r(V_r + B_S)^T) + U_{-r}S_{-r}V_{-r}^T$ has rank in $[n - 2r, n]$. Therefore, $\mathcal{R}(\text{LoRA}; W) = r$ and $\mathcal{R}(\text{Spectral Adapter}^A; W) = 2r$ can be derived trivially. □

## C  Cayley Parameterization Proof

*Proof.* With any trainable square matrix $A$, we set $Q = (A - A^T)/2$ and thus $Q = -Q^T$ and $Q$ is skew-symmetric thereby. Now we show that for any skew-symmetric $Q$, $(I + Q)(I - Q)^{-1}$ is orthogonal. Let $O = (I + Q)(I - Q)^{-1}$, then

$$
\begin{aligned}
O^T O &= ((I + Q)(I - Q)^{-1})^T (I + Q)(I - Q)^{-1} \\
&= (I - Q^T)^{-1}(I + Q^T)(I + Q)(I - Q)^{-1} \\
&\quad \text{by } Q \text{ skew-symmetric,} \\
&= (I + Q)^{-1}(I - Q)(I + Q)(I - Q)^{-1} \\
&\quad \text{since } (I - Q) \text{ and } (I + Q) \text{ have same eigen-basis and are commutable,} \\
&= I,
\end{aligned}
$$

which shows that the Cayley parametrization is exact and no re-SVD is needed for orthogonality preservation. □

## D   Connection to DoRA

In DoRA [32], the authors observe that plain LoRA method tends to either increase or decrease the magnitude and direction updates proportionally and thus lacks ability to make slight direction change together with large magnitude change, to come across this limitation, the authors propose to decompose pretrained model weights into magnitude and direction and update them separately. The magnitude is replaced with a trainable scalar and the direction is updated with original LoRA method. Experiments in [32] show that such decomposition helps improve effectiveness of LoRA significantly. Here we show that our Spectral Adapter$^A$ is closely connected to the weight decomposition trick used in DoRA when pretrained model weight is of vector form. We note that in DoRA, after the weight decomposition, each column becomes unit-length while in Spectral Adapter$^A$, we also operates on matrices with unit-length columns. Specifically, consider a pretrained model weight $w_0 \in \mathbb{R}^{n \times 1}$, then DoRA becomes

$$w = \underline{w} \frac{w_0 + \underline{ba}}{\|w_0 + \underline{ba}\|_2},$$

where $\underline{w}$ is a trainable scalar initialized at $\|w_0\|_2$. $\underline{b}$ and $\underline{a}$ are trainable parameters of size $n \times 1$ and $1 \times 1$ respectively, with $\underline{ba} = 0$ at initialization. Comparably, Spectral Adapter$^A$ becomes

$$w = \left(\frac{w_0}{\|w_0\|_2} + \underline{a}'\right)\|w_0\|_2(1 + \underline{b}'),$$

with trainable vector $\underline{a}' \in \mathbb{R}^{n \times 1}$ and trainable scalar $\underline{b}'$ both initialized at zero. We can thus equivalently view $\|w_0\|_2(1 + \underline{b}')$ as a single trainable scalar initialized at $\|w_0\|_2$, which then plays the role of magnitude adapter as $\underline{w}$ in DoRA. $\underline{a}'$ is adopted for directional adaptation since it directly operates on the normalized base vector.

## E   Cost Investigation (More Detailed)

Here we address the potential concern about the overhead of our proposed spectral adaptation mechanism. Firstly, we note that spectral adapter introduces similar number of trainable parameters and can be merged into original model weights, thus it is lightweight for sharing and introduces no additional inference latency, which preserves the strengths of additive fine-tuning methods. Therefore, the major overhead concern exists in the runtime and GPU storage overhead during online training. Note our method involves only matrix multiplication in the forward procedure and thus should run as quick as LoRA. Though the SVD procedure can bring additional runtime overhead, it needs to be done only once for a single model and can be reused for later fine-tuning on various downstream tasks. Besides, modern numerical tools such as randomized SVD [13] can also be exploited and the SVD procedure can be parallelized when multiple machines are available. As for GPU storage, unlike SVDiff [15] where all SVD components are required for training procedure thus introducing significant GPU storage burden, our method requires only the top spectral space to be stored additionally and consumes similar GPU storage to LoRA for relatively small tuning ranks (which is usually the case).

## F   Supplemental Materials for Experiments

### F.1   Experimental Setup for Figure 1

For Figure 1 experiments, we follow QDoRA [53] experimental setup for fine-tuning Llama3 8B model, where all k_proj, q_proj, v_proj, up_proj, down_proj, and gate_proj weights are tuned. We adopt the same data processing method and train on $10K$ Orca Math data (shuffled) as in [53]. We fix learning rate as $1e - 5$ for all methods as in QDoRA and train for one epoch with batch size $8$. $r = 8$ is adopted for LoRA, DoRA, AdaLoRA, and Spectral Adapter$^A$ while for OFT, we set number of diagonal blocks to be 800 to maintain similar amount of trainable parameters. LoRA alpha is set to be 16 following DoRA [32] convention and AdaLoRA hyperparameter is set following what has been used for MNLI benchmark in the original AdaLoRA report [65] with regularization set to $1e - 3$ which we find works better. For evaluation, we test on GSM8K [8] benchmark for exact matching. For more comparisons, Figure 9 provides training loss for smaller rank $r = 4$ (oft_$r$ = 1600) and larger rank $r = 64$ (oft_$r$ = 95). All settings are the same except that LoRA alpha is always kept as

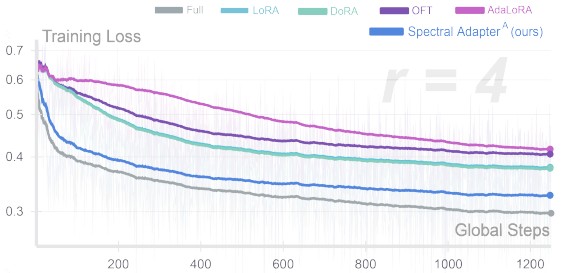 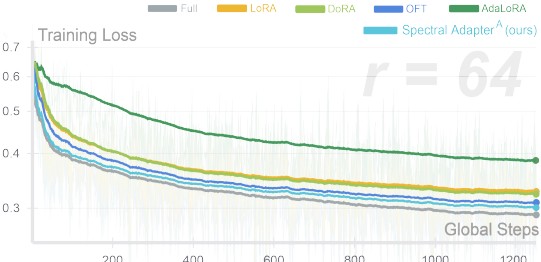

Figure 9: More experiments with Llama3 8B model with different number of trainable parameters. In the left plot, the training loss of LoRA and DoRA overlaps. See Appendix F.1 for details.

twice as rank number. From Figure 9 we can observe that though increasing trainable parameters closes the gap between different tuning methods, our spectral adapter method is always superior to other PEFT methods and stays closest to full fine-tuning.

## F.2 Hyperparameter Setting for DeBERTaV3-base Experiment (Section 4.1)

| Dataset | learning rate | batch size | #epochs | optimizer | weight decay |
|---------|---------------|------------|---------|-----------|--------------|
| MNLI | $1e-4$ | 32 | 1 | AdamW | 0.01 |
| RTE | $3e-4$ | 32 | 10 | AdamW | 0.01 |
| QNLI | $1e-4$ | 32 | 1 | AdamW | 0.01 |
| MRPC | $7e-4$ | 32 | 13 | AdamW | 0.01 |
| QQP | $1e-4$ | 32 | 10 | AdamW | 0.01 |
| SST-2 | $1e-4$ | 32 | 5 | AdamW | 0.01 |
| CoLA | $3e-4$ | 32 | 8 | AdamW | 0.01 |
| STS-B | $5e-4$ | 32 | 30 | AdamW | 0.01 |

Table 4: Hyperparameters for DeBERTaV3-base model fine-tuning with Spectral Adapter$^A$ in Section 4.1

Table 4 shows the hyperparameter setting for our Spectral Adapter$^A$ used for fine-tuning DeBERTaV3-base model in Section 4.1. We set number of diagonal blocks to be 4 and enable block sharing for OFT to maintain similar amount of trainable parameters.

## F.3 More About DeBERTaV3-base Experiment

Left plot in Figure 10 presents the training loss and validation score comparisons of LoRA, SVDiff and our Spectral Adapter$^A$ for fine-tuning DeBERTaV3-base model on CoLA benchmark. We set learning rates for both LoRA and Spectral Adapter$^A$ as what has been used in popular public blog [40] for LoRA fine-tuning with DeBERTaV3-base model, which is not tuned in favor of our method. For SVDiff, since it is originally proposed for vision model tuning, we extend it to this experiment by tuning all singular values of pretrained weights. We find the same learning rate leads to poor fine-tuning results with SVDiff, we thus pick the best learning rate among $[1e-3, 1e-4, 1e-5]$ according to validation performance and set learning rate to be $1e-3$. We use $r=8$ for LoRA and Spectral Adapter$^A$. From Figure 10, it can be observed that Spectral Adapter$^A$ achieves better training and validation performance compared to both LoRA and SVDiff.

Interestingly, in LoRA [20], the authors provide a correlation analysis between the LoRA additive component $\triangle W = AB^T$ and original pretrained weight matrix $W$ (see Section H.3 in [20]), and they find that the additive component does not contain the top singular directions of $W$. The authors therefore conclude that the learned LoRA component amplifies "task-specific" directions which are not emphasized in the pretrained weight matrix. Naively, this seems to suggest that tuning top singular subspace of pretrained weights is not ideal and one should identify the desired "task-specific" directions to improve LoRA. Here we show that this is not the case and fine-tuning top directions provides a significant improvement to LoRA. In the right plot of Figure 10 above, we experiment

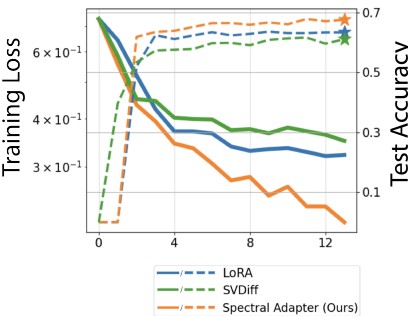 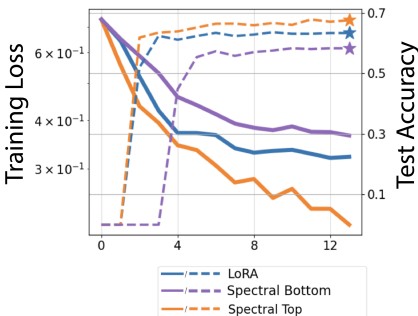

Figure 10: Left plot presents training loss and validation results for fine-tuning DeBERTaV3-base model with LoRA, SVDiff, and Spectral Adapter$^A$ on CoLA benchmark. Right plot compares the same statistics between LoRA and spectral adapter with top ranks and bottom ranks tuned respectively.

tuning the top eighth rank and the bottom eighth rank of singular vector space in our Spectral Adapter$^A$, which we present as "Spectral Top" and "Spectral Bottom" respectively. Remarkably, "Spectral Top" converges faster and scores higher than LoRA, which is then superior to "Spectral Bottom". This result unravels the fact that tuning different part of spectral space brings different tuning effect and tuning the top columns of singular vector space improves LoRA tuning significantly. See Section 3 for more theoretic insights.

### F.4 Hyperparameter Setting for Mistral 7B Experiment (Section 4.1)

| Method | lr | lora alpha | batch size | #epochs | lora dropout | weight decay |
|---|---|---|---|---|---|---|
| LoRA | $2.5e-5$ | 16 | 4 | 2 | 0.05 | 0.01 |
| DoRA | $2.5e-5$ | 16 | 4 | 2 | 0.05 | 0.01 |
| Spectral Adapter$^A$ | $2.5e-5$ | - | 4 | 2 | - | 0.01 |

Table 5: Hyperparameters for Mistral 7B model fine-tuning task in Section 4.1

Table 5 shows training hyperparameter setting for fine-tuning Mistral 7B model in Section 4.1. We train with bfloat16 precision and fine-tune all q_proj, k_proj, v_proj, o_proj, and gate_proj weights. We evaluate with lm-evaluation-harness [47]. Table 6 shows accuracy comparison of different tuning methods with learning rate $1e-5$. Our Spectral Adapter$^A$ still exceeds both LoRA and DoRA.

### F.5 Supplemental Materials for Multi-Adapter Fusion Experiment (Section 4.2)

#### F.5.1 Comparison of Single Object Generation

We present more experimental results to show that Spectral Adapter$^A$ with top ranks tuned behaves at least as good as LoRA with same parameter budget and is better than Orthogonal Adaptation [42], which is likely due to that Orthogonal Adaptation fixes LoRA parameter $B$ and thus has limited expressiveness. We also show that tuning bottom ranks in spectral adapter behaves worse than all other methods. Figure 11 shows generation results for custom toy concept tuning, where Orthogonal Adaptation and Spectral Adapter$^A$ (bottom) generate inaccurate happy-face octopus, sad-face octopus, and green tortoise. Figure 12 shows generation results for custom animal concept tuning, where Orthogonal Adaptation and Spectral Adapter$^A$ (bottom) sometimes miss first dog concept.

| Method | #Param | GSM8K |
|---|---|---|
| Pre-Trained | – | 38.82 |
| LoRA$_{r=8}$ | 0.16% | $43.29 \pm 1.36$ |
| DoRA$_{r=8}$ | 0.17% | $43.52 \pm 1.37$ |
| Spectral$^A_{r=8}$ | 0.16% | $46.47 \pm 1.37$ |

Table 6: Supplemental experiments of fine-tuning Mistral 7B model with different PEFT methods with a different learning rate on GSM8K benchmark. See Section F.4 for experimental details.

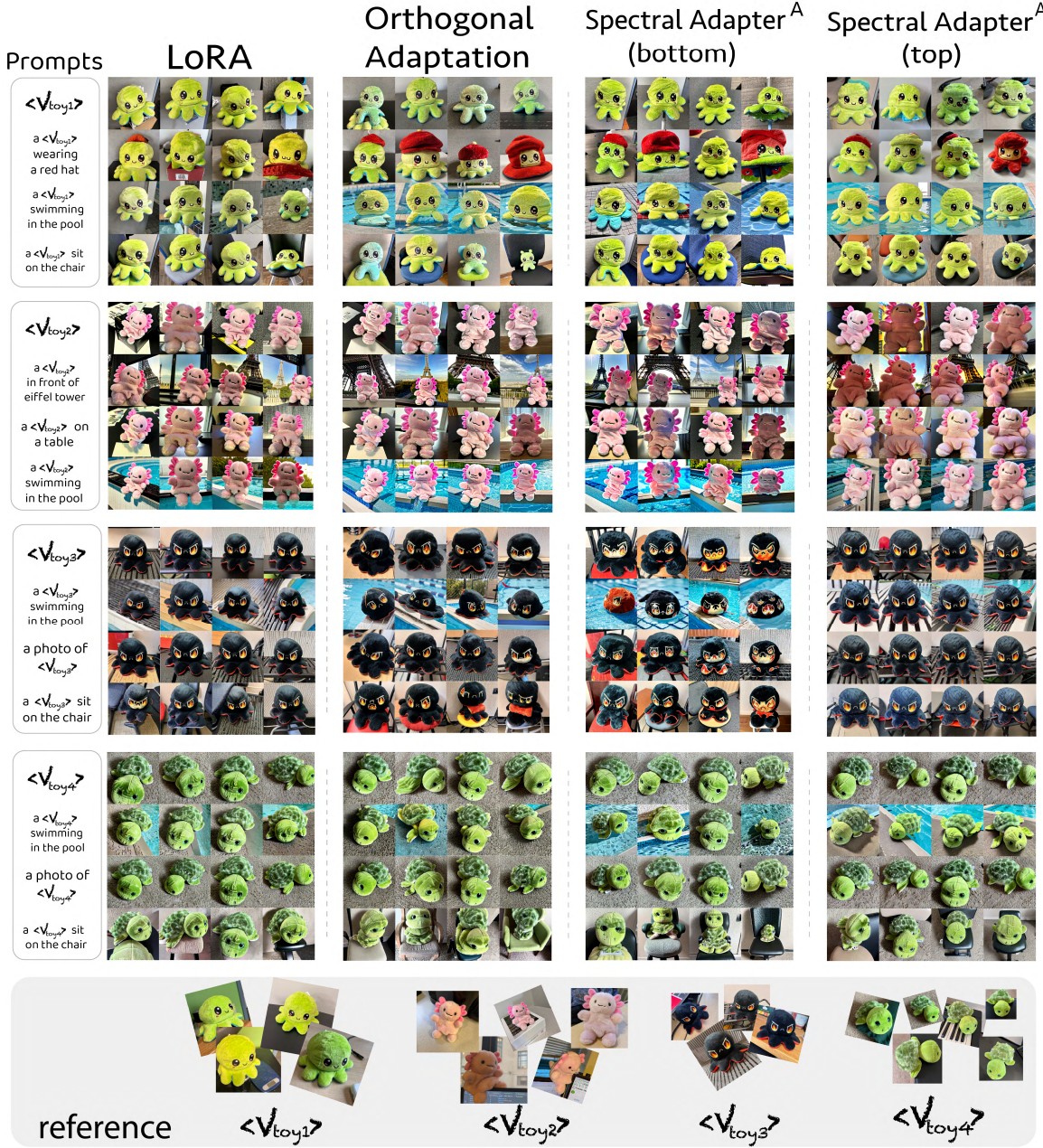

Figure 11: Generation results for single toy concept tuning with LoRA, Orthogonal Adaptation, and Spectral Adapter$^A$ with top and bottom ranks tuned respectively.

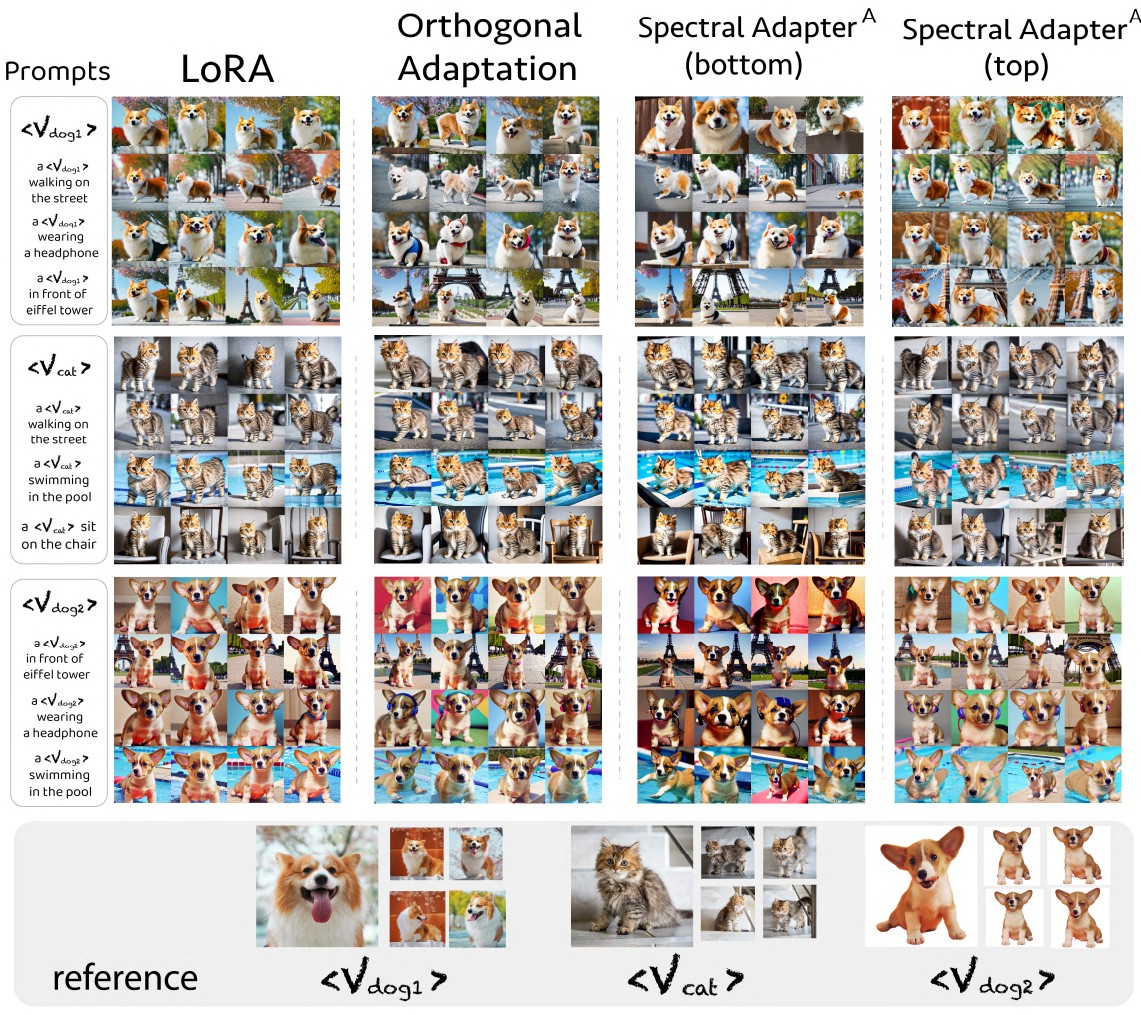

Figure 12: Generation results for single animal concept tuning with LoRA, Orthogonal Adaptation, and Spectral Adapter$^A$ with top and bottom ranks tuned respectively.

### F.5.2 More Multi-Adapter Fusion Generation Results

Here we present more results for multi-adapter fusion generation. Figure 13 shows generation results for multi-object generation for custom toy concepts and Figure 14 presents generation results for multi-character generation for three computer scientists. See below for experimental details.

**Multi-Object Generation.** As in Section 4.2, we fine-tune Chilloutmix diffusion model [1] on four custom toy concepts, see "reference" in Figure 13 for original toy images. We use $r = 8$ for all methods and tune first, second, third, and fourth top eighth columns of singular vector space of pretrained weights for first, second, third, and fourth toys in our Spectral Adapter$^A$. We follow all default experimental settings in [12] and tune all embedding layer, U-Net, and text-encoder. For better spatial alignment, we employ T2I-Adapter with sketch condition listed in "reference" in Figure 13. We randomly select three scenes and prompt fused-adapters for the results, see "prompts" in Figure 13 for individual prompt being used. From Figure 13, it can be observed that FedAvg and Orthogonal Adaptation generate unsatisfactory happy-face octopus and green tortoise toys. On the contrary, our spectral adapter generates high-quality images similar to Gradient Fusion while saving much more time.

**Multi-Character Generation.** We also experiment fine-tuning Chilloutmix diffusion model [1]

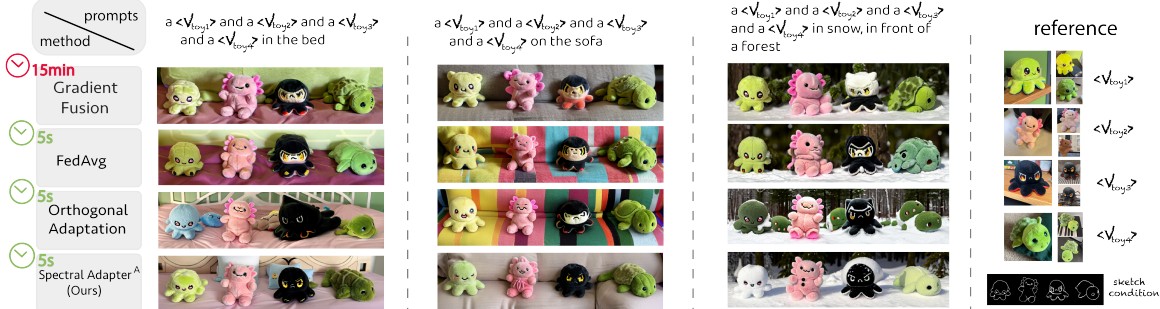

Figure 13: Generation results of Chilloutmix diffusion model [1] tuned on four custom toy concepts with different fused adapters. See Appendix F.5.2 for details.

with photos of three computer scientists Yoshua Bengio, Yann LeCun, and Geoffrey Hinton. As in multi-object generation, we use $r = 8$ for all methods and tune first, second, and third top eighth columns of singular vector space of pretrained weights for Bengio, Lecun, and Hinton in our Spectral Adapter$^A$. We use T2I-Adapter [39] with keypose condition. See "reference" in Figure 14 for scientists' photos and keypose condition being used. Figure 14 shows generation results for prompt "$<V_{\text{bengio}}>$ and $<V_{\text{lecun}}>$ and $<V_{\text{hinton}}>$, standing near a lake, 4K, high quality, high resolution" with different fused adapters, from which it can be observed that our spectral adapter generates picture of most consistent styles across characters and renders all scientists' faces clearly.

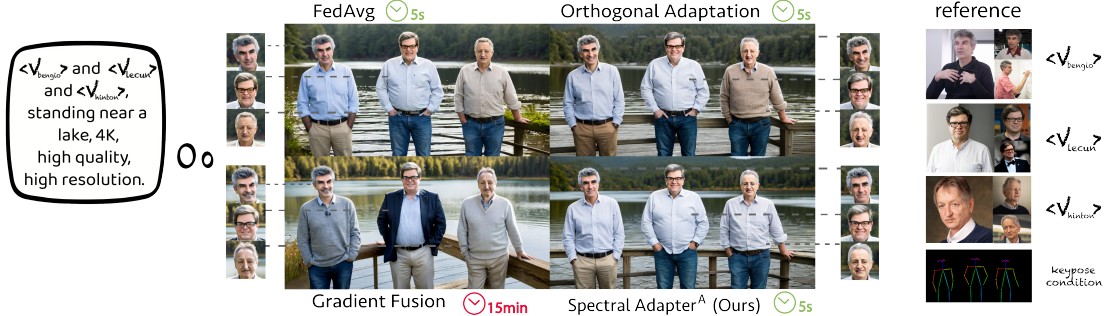

Figure 14: Generation results of Chilloutmix diffusion model [1] tuned on photos of three computer scientists with different fused adapters. See Appendix F.5.2 for details.

## F.6 Supplemental Materials for Parameter Efficiency Experiment (Section 4.3)

| Method | text encoder lr | unet lr |
|---|---|---|
| LoRA | $1e-5$ | $1e-4$ |
| VeRA ($r = 1$) | $1e-3$ | $1e-4$ |
| VeRA ($r = 1024, 4096$) | $5e-3$ | $1e-4$ |
| OFT$^A$ | $1e-5$ | $1e-4$ |
| LiDB | $5e-4$ | $1e-4$ |
| SVDiff | $1e-3$ | $1e-4$ |

Table 7: Hyperparameters for baseline methods for diffusion model fine-tuning task in Section 4.3

In this section, we present more tuning results with various parameter budgets for parameter efficiency experiment studied in Section 4.3, see Section 4.3 for baseline method explanation. Table 7 shows the learning rates used for each baseline method and Table 8 shows learning rates used for our method, the rest experimental settings are default as in [12].

| Method | vase | | chair | | table | |
|---|---|---|---|---|---|---|
| | text | unet | text | unet | text | unet |
| Spectral Adapter$^R$ ($r = 2, 40$) | $1e-3$ | $1e-2$ | $1e-2$ | $1e-2$ | $1e-3$ | $1e-2$ |
| Spectral Adapter$^R$ ($r = 4$) | | | $5e-3$ | $5e-3$ | $1e-3$ | $1e-2$ |
| Spectral Adapter$^R$ ($r = 8$) | $5e-4$ | $5e-2$ | $1e-3$ | $1e-2$ | $1e-3$ | $1e-2$ |
| Spectral Adapter$^R$ ($r = 16$) | | | $1e-2$ | $1e-3$ | $1e-3$ | $1e-2$ |
| Spectral Adapter$^R$ ($r = 24$) | $1e-4$ | $1e-2$ | $1e-3$ | $1e-3$ | $1e-4$ | $1e-2$ |
| Spectral Adapter$^R$ ($r = 32$) | $1e-4$ | $5e-2$ | | | | |

Table 8: Hyperparameters for Spectral Adapter$^R$ for diffusion model fine-tuning task in Section 4.3

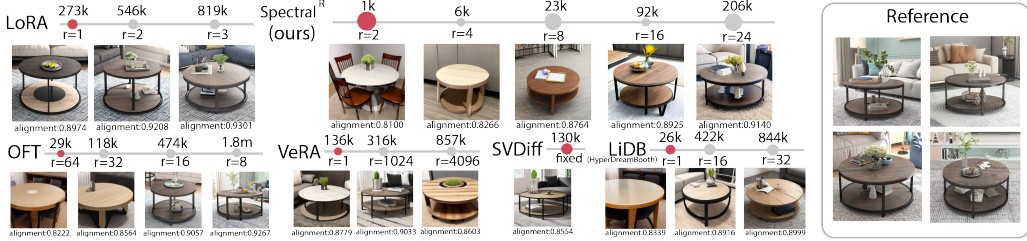

Figure 15: Generation results for prompt "a <$V_{\text{table}}$>" after fine-tuning Chilloutmix diffusion model [1] on custom table images with different PEFT methods. Spectral$^R$ is abbreviation for Spectral Adapter$^R$. See Appendix F.6 for details.

Figure 15 shows generation results of Chilloutmix diffusion model [1] fine-tuned on custom table concept with different methods under various parameter budgets. The prompt used is "a <$V_{\text{table}}$>". LoRA generates acceptable images for all rank $r = 1, 2, 3$ though it starts with $273k$ parameters even if rank is set to $1$. OFT generates desirable images only for parameter budget $> 400k$. VeRA and LiDB start to generate reasonable images with $> 300k$ trainable parameters and SVDiff has only a single fixed parameter budget. Meanwhile, our Spectral Adapter$^R$ recognizes the shape of custom table with as few as $6k$ parameters and produces ideal images since $100k$ parameters. See Appendix F.7 for alignment score computation details.

### F.7 Alignment Score Computation

For better quantitative measurement, we compute alignment scores for our Figure 5,6,7,15 results. Specifically, we first compute CLIP [46] embedding for all generated/reference images and prompt texts, then we compute the cosine similarity between generated images' embedding and reference images' embedding to serve as their alignment score. Likewise, text score stands for cosine similarity between generated images' embeddings and their corresponding prompt texts' embeddings. Intuition here is that if an image is close to another image (or text), their CLIP vectors are expected to stay close as well. For Figure 5 alignment score computation, we crop each generated image vertically into three columns, then we compute their alignment scores to each corresponding reference animal, we finally take the mean of these three scores. For Figure 6, 7, 15 scores, we compute average score over three random trials, with each trial consisting of $8$ generated images.

