# OpenReview forum: "Spectral Adapter: Fine-Tuning in Spectral Space"
_NeurIPS.cc/2024/Conference — NeurIPS 2024 poster_

### Official Review · Reviewer_qJVf · 2024-07-01

**Soundness:** 3
**Presentation:** 3
**Contribution:** 2
**Rating:** 6
**Confidence:** 4

**Summary:**

In summary, this paper investigates advancements in Parameter-Efficient Fine-Tuning (PEFT) for pre-trained neural networks by integrating spectral information from pretrained weights, aiming to enhance the classic LoRA approach. By employing Singular Value Decomposition (SVD), the authors introduce two spectral adaptation techniques: additive tuning and orthogonal rotation of the top singular vectors.

**Strengths:**

The topic of LoRA adaptation addressed in this paper is valuable due to its applicability to a wide range of high-level and low-level tasks. The authors introduce two variants of the proposed enhanced LoRA method. In specific datasets and tasks, these variants demonstrate improvements over state-of-the-art (SOTA) methods.

**Weaknesses:**

There are some typos that need to be corrected. For example, a spelling mistake in the word "Apppendix" in line 32, it should be corrected to "Appendix." Additionally, the word "digged" in line 20 should be corrected to "dug" to use the proper past tense of the verb "dig.".

It is not common to put the whole literature review in Appendix.

In line 53, the phrase "orthogonal rotating the top singular vector space." This phrase should be corrected to "orthogonally rotating the top singular vector space" to properly use the adverbial form "orthogonally," which modifies the verb "rotating."

Additionally, if the spectral space is modified, the rank will also change. This implies that the optimal ranks of the two spectral spaces differ, making it unfair to compare different LoRA absed methods using the same rank.

There is a logic error of the claim stated in line 82-93: Specifically, the statement that "these methods require storing all U, S and V during training while only the diagonal vector of S is tuned, which nearly doubles the storage requirement compared to pretraining when fine-tuning on downstream tasks" is misleading. Storing all components (U, S, and V) does indeed increase storage, but it's not clear why this would "nearly double" the storage requirement. The increase in storage would depend on the specifics of the matrix dimensions and the storage format. The phrase needs to clarify how the storage requirement nearly doubles to avoid logical inconsistency.

**Questions:**

Literature [36, 4, 56] already studied the  spectral space of model weights; it is not clear what is new in this paper, and highlighting the difference would make the contribution of this paper much clearer.

The inclusion of the revised Singular Value Decomposition (SVD) based fine-tuning, as compared to classic LoRA, is time-consuming and the computational burden increases with the number of fine-tuned blocks or layers. How does the paper address this issue?

**Limitations:**

Instead of fine-tuning the singular values of the weights, this paper proposes fine-tuning the singular vectors of the weights. However, the motivation behind this approach is not convincingly presented. For example, The motivation for fine-tuning the singular value vectors is not clearly articulated. As we know, U is an orthogonal matrix representing the left singular vectors, S is a diagonal matrix of singular values, and V is an orthogonal matrix representing the right singular vectors. A critical question that arises is whether U and V maintain their orthogonality after fine-tuning. Orthogonal U and V matrices provide an optimal basis for representing the weight matrices. Losing orthogonality results in a suboptimal basis, which can lead to less efficient representations of the neural network weights. Also, without orthogonality, the interpretability and distribution of the singular values will also be affected. Furthermore, orthogonal matrices are numerically stable and well-conditioned, meaning small changes in the data lead to small changes in the results. Without orthogonality, the resulting matrices may become ill-conditioned, causing numerical instability and issues in the training and optimization algorithms.

Additionally, the key contribution of the paper lacks clarity and contains logical errors. It would be helpful if the authors could address these concerns and provide further clarification.

---

> ### Author Rebuttal · Authors · 2024-08-06
>
> Thanks for the careful reviews. Given the rebuttal character limit, we address $\textbf{most important questions}$ here, followed by some $\textbf{other comments later}$.
>
> $\textbf{Weaknesses 4.}$ Additionally, if the spectral space is modified, the rank will also change. This implies that the optimal ranks of the two spectral spaces differ, making it unfair to compare different LoRA based methods using the same rank.
>
> $ \textbf{Answer to Weaknesses 4.}$ The general goal for parameter-efficient fine-tuning is simply to achieve better training loss (and hopefully validation loss as well) with fewer trainable parameters, and is thus not to decide what is the optimal rank and how the spectral space changes. Our proposed method shows better train/test performance, e.g., Figure 1; leads to better model fusion results, e.g., Figure 5; and is more parameter-efficient, e.g., Figure 6. This only requires one round of SVD of the weight matrices, which induces ignorable overhead, e.g., Figure 7.
>
> Moreover, we are not simply comparing with LoRA or LoRA-based models, we compare with state-of-the-art fine-tuning models including OFT which is less like LoRA, and another spectral adaptation mechanism SVDiff. See Figure 6 for example.
>
> We acknowledge it's possible that optimal ranks for different fine-tuning models may differ, thus we test over different ranks. See Figure 1 and 8 for results with rank 4, 8, and 64, which are most commonly used ranks in nowadays fine-tuning tasks. Our method is better than LoRA/LoRA-based/other methods for all cases.
>
> $\textbf{Weaknesses 5.}$. There is a logic error of the claim stated in line 82-93: Specifically, the statement that "these methods require storing all U, S and V during training while only the diagonal vector of S is tuned, which nearly doubles the storage requirement compared to pretraining when fine-tuning on downstream tasks" is misleading. Storing all components (U, S, and V) does indeed increase storage, but it's not clear why this would "nearly double" the storage requirement. The increase in storage would depend on the specifics of the matrix dimensions and the storage format. The phrase needs to clarify how the storage requirement nearly doubles to avoid logical inconsistency.
>
>
> $ \textbf{Answer to Weaknesses 5.}$ For simple demonstration, assume we have a weight matrix $W\in\mathbb{R}^{n\times n}$ which is of full rank. Then after the singular value decomposition, we will get $U\in\mathbb{R}^{n\times n}, S\in\mathbb{R}^n,V\in\mathbb{R}^{n\times n}.$ SVDiff proposes to tune all singular values, thus it requires to store all $U,S,V$ in their full format, which results in $(2n^2+n)\*$sizeof(float) storage overhead plus $n$ trainable parameters. On the contrary, consider our spectral adapter which tunes only the top-$r$ columns of $U$ and $V$, thus we form and store $W_2=U[r:]\text{diag}(S[r:])V[r:]^T$  together with top-$r$ columns of $U,V$ and top-$r$ elements of $S$. In our training, $W_2$ can be summed up with the top spectral part. This paradigm results in $(n^2+2rn+r)\*$sizeof(float) storage overhead with additional $2nr$ trainable parameters for additive spectral adapter and $2r^2$ trainable parameters for rotational spectral adapter. Since $n$ is usually much larger than $r$ in training PEFT models, SVDiff requires $\sim 2n^2*$sizeof(float) storage, which doubles  $\sim n^2*$sizeof(float) storage needed by our spectral adapter. This storage explosion is also observed in our empirical  training of SVDiff compared to spectral adapter.
>
> $\textbf{Questions 1.}$  Literature [36, 4, 56] already studied the spectral space of model weights; it is not clear what is new in this paper, and highlighting the difference would make the contribution of this paper much clearer.
>
> $ \textbf{Answer to Questions 1.}$  [36] studies specifically the singular value distribution of weight matrices of LLM,  it is more of a theoretic work in field of statistical ML. The conclusion there is that singular value distribution is more structured for larger model and varies across different training phases. [56] has nothing to do with spectral space of weight matrices, it only observes that strong attention scores are often attached to initial tokens in LLM training, which they dub as "attention sink". [4] establishes connection between attention sink and spectral space of weight matrices, it suggests that the bottom spectral space of NN weights plays a critical role in attention sink phenomenon. None of this work proposes to exploit spectral space of NN weights in fine-tuning tasks. Thus the difference between these works and ours is huge.
>
> $\textbf{Questions 2.}$ The inclusion of the revised Singular Value Decomposition (SVD) based fine-tuning, as compared to classic LoRA, is time-consuming and the computational burden increases with the number of fine-tuned blocks or layers. How does the paper address this issue?
>
> $\textbf{Answer to Questions 2.}$ We note that the overhead brought by computing SVD of the pretrained weights is one-round and can be cached. After the matrix decomposition is obtained and stored, online training would induce similar burden as classic LoRA setting. To further investigate the practicality of our proposed method, we involve in Section 4.4 both the runtime and storage comparison of our method compared to LoRA. With large models of current size, the induced runtime overhead of the SVD procedure is marginal compared to the training time used. Meanwhile, the GPU storage needed by our method is close to LoRA. We have also discussed about both online and offline training and testing cost in Appendix E.

---

> ### Author Response · Authors · 2024-08-06
>
> $\textbf{Weaknesses 1.}$ There are some typos that need to be corrected. For example, a spelling mistake in the word "Apppendix" in line 32, it should be corrected to "Appendix." Additionally, the word "digged" in line 20 should be corrected to "dug" to use the proper past tense of the verb "dig".
>
> $\textbf{Answer to Weaknesses 1.}$ Thanks for pointing out these issues, we've corrected the spelling and the tense of the word in our revision.
>
> $\textbf{Weaknesses 2.}$ It is not common to put the whole literature review in Appendix.
>
> $\textbf{Answer to Weaknesses 2.}$  The decision of putting literature review into appendix is due to the page limit, we will move it into the main content with more page budget in camera ready version if our work gets accepted. Moreover, since we already involve baseline method explanation and all its relative properties  whenever a prior method is compared to, e.g., bottom part of page 6 and middle part of page 8 (with Table 3 summarizing the key features), we believe that postponing literature review into appendix doesn't affect reading and understanding. Also, our introduction part (Section 1) and methodology part (Section 2)  already draw connection to most relevant work, the literature review section (Appendix A) includes additional works that are less relevant and is presented for completeness and credit for all related work in related research fields.
>
> $\textbf{Weaknesses 3.}$ In line 53, the phrase "orthogonal rotating the top singular vector space." This phrase should be corrected to "orthogonally rotating the top singular vector space" to properly use the adverbial form "orthogonally," which modifies the verb "rotating."
>
> $\textbf{Answer to Weaknesses 3.}$ Thanks for the suggestion, we've corrected the phrase in our revision.
>
> $\textbf{Limitations.}$ Instead of fine-tuning the singular values of the weights, this paper proposes fine-tuning the singular vectors of the weights. However, the motivation behind this approach is not convincingly presented. For example, The motivation for fine-tuning the singular value vectors is not clearly articulated. As we know, U is an orthogonal matrix representing the left singular vectors, S is a diagonal matrix of singular values, and V is an orthogonal matrix representing the right singular vectors. A critical question that arises is whether U and V maintain their orthogonality after fine-tuning. Orthogonal U and V matrices provide an optimal basis for representing the weight matrices. Losing orthogonality results in a suboptimal basis, which can lead to less efficient representations of the neural network weights. Also, without orthogonality, the interpretability and distribution of the singular values will also be affected. Furthermore, orthogonal matrices are numerically stable and well-conditioned, meaning small changes in the data lead to small changes in the results. Without orthogonality, the resulting matrices may become ill-conditioned, causing numerical instability and issues in the training and optimization algorithms.
>
> $\textbf{Answer to Limitations.}$ Thanks for raising this point. In our paper, we propose two spectral fine-tuning paradigms: an additive version and a rotational version. Our Spectral Adapter$^R$ applies a trainable rotation to the top-$r$ left and right singular vectors and preserves the orthogonality. This is achieved by using the differentiable Cayley parameterization. Our
> Lemma 4.1 proves that Spectral Adapter$^R$ with Cayley parameterization maintains orthogonality of the singular vectors. This follows by parameterizing a skew-symmetric matrix $Q=(A-A^T)/2$ and letting the rotation matrix be $(I+Q)(I-Q)^{-1}$. In Appendix C, we provide a proof that $(I+Q)(I-Q)^{-1}$ is orthogonal.
>
> On the other hand, our second version, Spectral Adapter$^A$, drops the orthogonality constraint. However, after fine-tuning, the weights are close to orthogonal since only the top-$r$ gets updated and the gradient updates are of small magnitude. We find in our experiments that both adapters work very well, and outperform low rank adapters. In this version, the reviewer is correct that dropping orthogonality will invalidate the optimality of the SVD decomposition in low-rank matrix approximation problems. However, in fine-tuning foundational models, the requirement of orthogonality of the fine-tuned weights is hard to justify. Meanwhile, we agree with the reviewer that interpretability of the fine-tuned model is improved when we use orthogonal fine-tuned weights. This is provided by our Spectral Adapter$^R$. If one wants to perform interpretability tasks which require singular values with our Spectral Adapter$^A$, just re-SVD the tuned matrix and continue analysis with the new set of basis.

---

> > ### Comment · Reviewer_qJVf · 2024-08-11
> > **Response to author rebuttal**
> >
> > The reviewer has carefully examined both the authors' rebuttal and the feedback from other reviewers. While many of the concerns have been addressed satisfactorily, the implementation of the revised Singular Value Decomposition (SVD)-based fine-tuning method, in contrast to the traditional LoRA approach, remains time-intensive. Furthermore, the computational load escalates with each additional fine-tuned block or layer. The authors have not proposed a solution to this issue in the current version. However, the overall quality of the manuscript has indeed improved post-rebuttal compared to the initial submission. Consequently, the reviewer has decided to increase the initial evaluation score.

---

> ### Author Response · Authors · 2024-08-06
>
> Please let us know whether our replies address all your concerns.  If so, could you please kindly consider increasing the score? If not, we are willing to address any other question in more details. Thanks!

---

### Official Review · Reviewer_7N9o · 2024-07-09

**Soundness:** 3
**Presentation:** 2
**Contribution:** 3
**Rating:** 5
**Confidence:** 5

**Summary:**

The paper proposes fine-tuning pretrained model weights in the spectral space for parameter efficiency. It explores two spectral adaptation mechanisms: additive tuning and orthogonal rotation of top singular vectors. The authors introduce these methods, providing theoretical analyses on rank capacity and robustness to support their approach. Experiments on language and diffusion model fine-tuning demonstrate the proposed method's superiority over previous parameter-efficient fine-tuning techniques.

**Strengths:**

1. The proposed spectral adapters, which introduce spectral adaptation, are interesting.

2. The theoretical analysis showing that spectral adapters have a larger rank capacity than LoRA is reasonable.

3. Experiments on language and diffusion model fine-tuning demonstrate that the proposed method outperforms other parameter-efficient fine-tuning techniques while maintaining efficiency.

**Weaknesses:**

1. The analysis of spectral adaptation robustness in Section 3.2 could be clearer. It would be helpful to provide what $\mathcal{R}(X)$ denotes and more clearly explain why fine-tuning $u^*$ is considered noiseless.

2. The paper's organization might benefit from some restructuring. Consider moving certain content, such as Lemma 4.1 and Table 3, from the experiments section to the methods section. Additionally, restructuring the experiments section could improve clarity.

3. While spectral adaptation is proposed, it would be valuable to more clearly demonstrate in the methods section how this approach is superior to prior works like SVDiff and OFT.

4. In Figures 1 and 8, using validation loss instead of training loss only for comparing PEFT methods could provide more meaningful insights.

5. It might be beneficial to discuss limitations in a separate section rather than within the Experiments section.

6. Including quantitative measures alongside the qualitative comparisons in the image generation results could strengthen the analysis.

**Questions:**

N/A

**Limitations:**

The authors mention some limitations in the checklist part.

---

> ### Author Rebuttal · Authors · 2024-08-06
>
> Thanks for the careful reviews. Given the rebuttal character limit, we address $\textbf{most important questions}$ here, followed by some $\textbf{other comments later}$.
>
> $\textbf{Weaknesses 1.}$  The analysis of spectral adaptation robustness in Section 3.2 could be clearer. It would be helpful to provide what $\mathcal R(X)$ denotes and more clearly explain why fine-tuning $u^\ast$ is considered noiseless.
>
> $\textbf{Answer to Weaknesses 1.}$ Sorry about the confusion, here $\mathcal R(X)$ denotes  the row space of data matrix $X$, i.e., $\mathcal R(X):=\text{row}(X).$ We've added the definition in our revision.
> By "noiseless'', what we mean here is that, we know from the theory that optimal weights need to lie in a certain subspace  for the toy example considered in Section 3.2. However, in practice they can deviate due to optimization errors. We find this subspace via SVD, effectively denoising the weights, and perform fine-tuning of this subspace. To explain more about the math derivation in Section 3.2, we provide a more concrete example in the official comment below.
>
> $\textbf{Weaknesses 4.}$ In Figures 1 and 8, using validation loss instead of training loss only for comparing PEFT methods could provide more meaningful insights.
>
> $\textbf{Answer to Weaknesses 4.}$ In Figure 1, we provide the training loss plot along with the test accuracy plot on the right panel (which is evaluated on the GSM8K benchbark measured using lm evaluation harness https://github.com/EleutherAI/lm-evaluation-harness). It can be seen that spectral adapter provides lower training loss and higher test accuracy. For Figure 8, since the reviewer raised this point, we have added similar test scores in our ``one-page attached pdf (left two panels in Section B)``. We have also included these new figures in our revision.
>
> $\textbf{Weaknesses 5.}$ It might be beneficial to discuss limitations in a separate section rather than within the Experiments section.
>
> $\textbf{Answer to Weaknesses 5.}$. Thanks for the suggestion, we will add an additional section for limitations in our revision. The main potential limitation of the proposed method is the overhead for computing SVD of weight matrices. However, for most modern networks (Stable Diffusion, Llama 3 and Mistral), this is tractable since the dimensions of layer weights are moderate. See Figure 7 which shows the SVD time/memory overhead is negligible. For certain other models with larger weights, our method may require more resources.
>
>  Another limitation to our discussion is that we only focus on tuning top spectral space, though we investigated tuning other spectral space sporadically, i.e., see Figure 9, 10, 11, Section 4.2, and ``Section B (right most plot) of our added one-page pdf``. However, there might be some cases where minor spectral space should be changed (as pointed out by reviewer Nthn) and the top spectral space might also shift during training (as pointed out by reviewer N6x8). Though the proposed method empirically works well, more advanced spectral space scheduling technique is worth exploring.
>
>
> $\textbf{Weaknesses 6.}$ Including quantitative measures alongside the qualitative comparisons in the image generation results could strengthen the analysis.
>
> $\textbf{Answer to Weaknesses 6.}$ Thanks for the suggestion.  To consolidate our work further, we have provided quantitative measurements for diffusion model results in our attached one-page pdf, and we have also appended these new results in a revision of our current paper. For the evaluation, we follow the same computation metrics as used in paper [12] and paper [42], where the cosine similarity between clip vectors are taken for distance measurement.  Specifically, in ``Section A.1 in that pdf``, we provide quantitative results for our Figure 5. For text alignment score, we compute with the following prompts corresponding to each column:
>
>  1) "two dogs and a cat in front of Mount Fuji",
>
>  2) "two dogs and a cat in a galaxy",
>
>  3) "two dogs and a cat on a playground".
>
>  For image alignment score, since we are generating multi-character plots, we crop each generation vertically into three parts, with each cropped plot containing only a single animal. We then compute the alignment score of each  cropped component with corresponding reference images of the same animal.  For generation of column 1, FedAvg scores best but has only 0.0005 higher average than our method. For generation of column 2 and 3, our method achieves highest average and is around 0.01 better than other methods.
>
>
>  ``Section A.2 of the attached pdf`` contains quantitative results for our Figure 6. We include both the trend curve for alignment score vs. parameter budget (left panel), and also the exact score number (right panel, please expand to see the number). We also shade the region with trainable parameter budget only achievable by our method. Notably, it can be clearly observed that our method already generates sensible images with very few parameters ($\leq$ 20K) that can never be  obtained by other methods. For larger parameter budget, our method is able to generate images of quality  comparable to SOTA. This exemplifies the parameter efficiency of our method.

---

> ### Author Response · Authors · 2024-08-06
>
> $\textbf{Additional Answer to Weaknesses 1.}$ consider the same two-layer ReLU model trained for minimizing squared loss, i.e.,
>
> $\qquad\qquad\qquad\qquad\qquad \\min_{W^{(1)},W^{(2)}} \\|(XW^{(1)})_+W^{(2)}-y\\|_2^2+\\beta(\\|W^{(1)}\\|_F^2+\\|W^{(2)}\\|_2^2$,
>
> where $X\in\mathbb{R}^{n\times d},W^{(1)}\in\mathbb{R}^{d\times m},W^{(2)}\in\mathbb{R}^{m}, y\in\mathbb{R}^n.$ Now we decompose each first-layer neuron $W_j^{(1)}\in\mathbb R^d$ (there are in total $m$ of them) into two parts $W_j^{(1)}=w_{j1}+w_{j2}$ where $w_{j1}$ lies in row space of $X$ and $w_{j2}$ is perpendicular to row space of $X$, i.e., $w_{j1}\in\mathcal R(X)$ and $w_{j2}\perp\mathcal R(X).$ This indicates $Xw_{j2}=0$ and thus $w_{j2}$ has no contribution to reduce the first quadratic loss term above. Since we have a second non-negative weight decay term, which would then result in  $w_{j2}=0$ when trained to optimality.
>
>  Therefore, when minimal loss is achieved, all first-layer neurons should lie in the row space of $X$, i.e., $W_j^{(1)}=w_{j1}\in\mathcal R(X),~\forall j.$  However, it might happen that due to optimization errors, some neurons are not exactly aligned with row space of $X$ and may have small value in the perpendicular direction. For demonstration, consider a toy example where we have two pieces of data, each of dimension three, i.e., $X\in\mathbb R^{2\times 3}$, and takes value
>  $$X=\begin{bmatrix}
>  1 & 0 & 0 \\\\
>  0 & 1 & 0
>  \end{bmatrix}.$$
>  Consider the case when we have three neurons, with first and second neurons being in row space of $X$ and third neuron perpendicular to row space of $X$ with a small value:
>  $$
>  W = \begin{bmatrix}
>  5 & 0 & 0 \\\\
>  0 & 7 & 0 \\\\
>  0 & 0 & 0.1
>  \end{bmatrix}.
>  $$
>  Then, using torch.svd on $W$ results in
>  $$
>  U = \begin{bmatrix} 0 & 1 & 0\\\\ 1 & 0 & 0\\\\ 0 & 0 & 1 \end{bmatrix}, S = \begin{bmatrix}7 & 5 & 0.1\end{bmatrix}, V=\begin{bmatrix}0 & 1 & 0\\\\ 1 & 0 & 0 \\\\ 0& 0 & 1 \end{bmatrix}.
>  $$
>  The top, second, and third spectral decomposition components would be
>  $$\\text{top:} \begin{bmatrix}0 & 1 & 0\end{bmatrix} \begin{bmatrix}7\end{bmatrix} \begin{bmatrix}0\\\\1\\\\0\end{bmatrix}=\begin{bmatrix}0 & 0 & 0\\\\ 0 & 7 & 0\\\\ 0& 0 & 0\end{bmatrix}, \text{second:}\begin{bmatrix}1 & 0 & 0\end{bmatrix}\begin{bmatrix}5\end{bmatrix}\begin{bmatrix}1\\\\0\\\\0\end{bmatrix}=\begin{bmatrix}5 & 0 & 0\\\\ 0 & 0 & 0\\\\ 0& 0 & 0\end{bmatrix}, \text{third:}\begin{bmatrix}0 & 0 & 1\end{bmatrix}\begin{bmatrix}0.1\end{bmatrix}\begin{bmatrix}0\\\\0\\\\1\end{bmatrix}=\begin{bmatrix}0 & 0 & 0\\\\ 0 & 0 & 0\\\\ 0& 0 & 0.1\end{bmatrix}.$$
>
> Therefore, if we focus on tuning the top two directions, we only need to deal with the ones aligned with row space of $X$ and are thus protected against small optimization errors, which we consider to be more robust and we address as "noiseless''.
>
>
> $\textbf{Weaknesses 2.}$ The paper's organization might benefit from some restructuring. Consider moving certain content, such as Lemma 4.1 and Table 3, from the experiments section to the methods section. Additionally, restructuring the experiments section could improve clarity.
>
> $\textbf{Answer to Weaknesses 2.}$ Thanks for the suggestions, our current structure follows: methodology (Section 2), theory (Section 3), experiments (Section 4), where the experiment part is splitted further into: language model (Section 4.1), diffusion model with additive spectral adapter (Section 4.2),  diffusion model with rotational spectral adapter (Section 4.3), runtime comparison (Section 4.4). We've tried hard on optimizing the layout of the paper and we want to keep methodology part (Section 2) short since we want to demonstrate what are the newly proposed method more clearly and succinctly. We have separate baseline discussions in Section 4.2 (bottom of page 6) and Section 4.3 (middle of page 8, including Table 3) since we are comparing with two different sets of baseline methods, that's why we haven't put Table 3 in the methodology section. The divergence in baselines being compared is because we are studying different characteristics of the fine-tuning model, i.e., we focus on model fusion in Section 4.2 and parameter efficiency in Section 4.3, thus we compare with different prior methods which are more proficient with respect to each of the aspect.
>
> Lemma 4.1 is more about methodology, but our primary concern is that it's specific to rotational adapter thus we put it in Section 4.3. We'll consider moving this part to the methodology section or we'll create another separate implementation section to discuss this implementation detail.
>
> We thank the reviewer again for bringing up the structure issue which suggests that our current paper layout might still cause come confusion, we'll do more optimization on this in our revision.

---

> ### Author Response · Authors · 2024-08-06
>
> $\textbf{Weaknesses 3.}$ While spectral adaptation is proposed, it would be valuable to more clearly demonstrate in the methods section how this approach is superior to prior works like SVDiff and OFT.
>
> $\textbf{Answer to Weaknesses 3.}$  Comparison/Advantage to SVDiff: idea wise, SVDiff proposes only to tune singular values, which is more constrained compared to our proposed method that considers tuning singular vector space. To see this, first of all, tuning singular values can be achieved by tuning singular vector space (just tune the scale of each row/column pair in $U$ and $V$); Secondly, tuning singular values would result in the same row/column space while tuning singular vector space has more freedom.  Practically, SVDiff requires to store all $U,S,V$ in their full format while our spectral adapter only tunes the top spectral component, thus the bottom part can be merged. For example, consider a weight matrix of dimension $n\times n$ with full rank, we have $U\in\mathbb R^{n\times n}, S\in\mathbb R^n, V\in\mathbb R^{n\times n}$ after singular value decomposition. Then SVDiff requires storing all $U,S,V$ and thus takes $(2n^2+n)\*$sizeof(float) GPU storage. For our spectral adapter, say we are tuning the top-$r$ spectral components, then we only need to store top-$r$ components of $U, S, V$  and we merge the rest as $W_2=U[r:]S[r:]V[r:]^T$ that we can sum up with the top part. Thus our method requires $(n^2+2rn+r)\*$sizeof(float) storage, which is two times smaller than SVDiff. Note we can not reduce beyond $n^2*$sizeof(float) since that is required to store the original weight matrix.
>
> Comparison/Advantage to OFT: OFT proposes to tune the row space of weight matrix by rotating them orthogonally. This is connected to our rotational spectral adapter which rotates the top columns of singular vector matrices. Intuitively, OFT is preserving the distance between neurons while our rotational spectral adapter is preserving the orthogonality of singular vector basis. These are two lines of logic thus it's hard to say our method is strictly superior to OFT in a rigorous sense. However, our method does have advantage in how it tackles the parameter efficiency issue. Vanilla OFT  method is not parameter-efficient since for any weight matrix $W\in\mathbb R^{n\times n}$, it needs to have another orthogonal permutation matrix $A\in\mathbb R^{n\times n}$ of the same size and it uses $AW$ in place of $W$. In OFT paper,  the authors  manually inject parameter efficiency by constraining $A$ to be only block diagonal with orthogonal blocks, and the authors claim that this constraint is not affecting much in their empirical experiments. Moreover, with such block diagonal $A$, the trainable parameter budget is constrained to be $r^2\*(n/r)$ where we assume the block size is $r\times r$ and it requires that $r|n$. On the contrary, our rotational spectral adapter orthogonally rotates the top $r$ columns of $U$ and $V$ matrices thus we are guaranteed with more parameter budget choices, i.e. our $r$ can take any value between $1$ and $n$, and our method is naturally parameter-efficient.

---

> ### Author Response · Authors · 2024-08-06
>
> We hope that our response has positively influenced your perception of our work. Please let us know if your queries have been addressed satisfactorily.  If so, could you please kindly consider increasing the score? If you require further clarifications of any points, we are enthusiastic about engaging in further discussion. Please do not hesitate to contact us. We highly value the generous contribution of your time to review our paper. Thanks!

---

> > ### Author Response · Authors · 2024-08-12
> >
> > Dear reviewer, since there are only two days left for author-reviewer discussion, we would like to confirm whether our responses have effectively addressed your concerns. We provided detailed responses to your concerns a few days ago, and we hope they have adequately addressed your issues. If you require further clarification on any of these points, please do not hesitate to contact us. Thanks!

---

> > > ### Comment · Reviewer_7N9o · 2024-08-12
> > >
> > > Thank you for the detailed rebuttal. After reviewing it, all my concerns have been addressed, so I am raising my score to 5.

---

### Official Review · Reviewer_N6x8 · 2024-07-12

**Soundness:** 3
**Presentation:** 2
**Contribution:** 3
**Rating:** 5
**Confidence:** 4

**Summary:**

In this paper, the authors proposed to modulate top-r singular vectors after performing SVD on the pretrained weights. Both theoretical analysis and experiments have shown that the proposed two types of spectral fine-tuning methods can improve the representation capacity of low-rank adapters.

**Strengths:**

1. The introduction of the proposed method is pretty clear, and the theoretical analysis and review of other PEFT methods help the readers quickly and comprehensively understand the core of the proposed method.
2. The advantages of the proposed method under low-rank conditions are very obvious (Figure 1 and Figure 8).

**Weaknesses:**

1. The proposed method have two versions, including spectral adapter^A and spectral adapter^R. However, the application boundaries of these two methods are not clear. Some experiments are applied by adapter^A, but others are applied by adapter^R. It's better to further discuss the difference or relationship between these two types of versions.
2. It's better to study the selection of columns of U and V. For example, bottom-r selection and random-r selection.
3. If we can combine addition and rotation?

**Questions:**

My main concerns are listed in the weaknesses.

Additional question: as illustrated in the paper, the spectral adapters are initialized once at the beginning. This is done by carrying out SVD on pre-trained model weights, identifying the most significant components to finetune on. However, analysis from GaLoRE (GaLore: Memory-Efficient LLM Training by Gradient Low-Rank Projection) shows that the primary components could shift from one to another during the training. So, it’s better to examine the possibility of re-parametrizing the spectral adapters back into model weights, doing SVD again, and creating new spectral adapters to catch new primary components periodically during the training. By doing this we might come up with a better result.

---

> ### Author Rebuttal · Authors · 2024-08-06
>
> Thanks for the careful reviews. Here are our answers to the questions.
>
> $\textbf{Weaknesses 1.}$ The proposed method have two versions, including spectral adapter-A and spectral adapter-R. However, the application boundaries of these two methods are not clear. Some experiments are applied by adapter-A, but others are applied by adapter-R. It's better to further discuss the difference or relationship between these two types of versions.
>
> $\textbf{Answer to Weaknesses 1.}$ Thanks for the suggestion, which is very reasonable. Our language model experiments are all done with additive spectral adapter. We carry out diffusion model experiments in Section 4.2 with additive spectral adapter and in Section 4.3 with rotational spectral adapter. These two sections are devoted to studying their different characteristics: in Section 4.2, we investigate the performance of additive spectral adapter for adapter fusion task. For different fine-tuning tasks, we propose to tune different spectral vector basis, which improves the generation results after they are merged. In Section 4.3, we explore the parameter efficiency capacity for rotational spectral adapter. Notably, the trainable parameter in rotational spectral adapter is only of size $\mathcal O(r^2)$ and thus can be very small for small $r.$ As far as we know, this is the only PEFT method that has trainable parameter budget scales only with $r$. Prior models which focus solely on reducing number of trainable parameters such as VeRA and LiDB (see Table 3) still have parameter budget scales with weight size. Remarkably, for only $r=2$, i.e., each weight matrix only induces $2*2^2=8$ trainable parameters, the fine-tuning already takes effect in some cases, e.g., Figure 15. No previous models can achieve this number of trainable parameter budget.
>
>
> Overall, we suggest additive spectral adapter for both general case fine-tuning and adapter fusion tasks, we suggest rotational spectral adapter as a remedy when there is strict constraint on trainable parameter budget. Thanks for raising this confusion, we will make the distinction clearer in our revision.
>
> $\textbf{Weaknesses 2.}$ It's better to study the selection of columns of U and V. For example, bottom-r selection and random-r selection.
>
> $\textbf{Answer to Weaknesses 2.}$  Thanks for the suggestion. We have included diffusion model results for bottom-$r$ selection (right plot of Figure 9, third columns of Figure 10 and Figure 11), which behaves much worse than top-$r$ selection and original LoRA model. We will consider moving some of them to the main text for better comparison.
>
> Since the reviewer has raised this point, we also do some additional experiments with top-$r$, middle-$r$, and bottom-$r$ tunings. See the third panel of ``Section B in our attached one-page pdf``. This experiment is for fine-tuning Llama 3 8B model (same as Figure 1) with $r=4$. Here for middle-$r$ tuning, we start at the $20$th column and tune consecutive $4$ columns beginning there, i.e., we are tuning the $20$th$\sim 24$th columns of $U$ and $V$. The result shows that tuning top-$r$ is better than middle-$r$, which is then better than bottom-$r$. One future direction is to study more random-$r$ selection techniques and perhaps dynamically varying the choice of $r$ based on some criterion, which would be in spirit close to AdaLoRA [64].
>
> $\textbf{Weaknesses 3.}$ If we can combine addition and rotation?
>
> $\textbf{Answer to Weaknesses 3.}$ Technically yes, but then it means one would need to store a set of heterogeneous fine-tuning models and need to distinguish them in forward pass since different merging methods need to be adopted. Thus it might bring some cumbersomeness.
>
> $\textbf{Questions.}$ as illustrated in the paper, the spectral adapters are initialized once at the beginning. This is done by carrying out SVD on pre-trained model weights, identifying the most significant components to finetune on. However, analysis from GaLoRE (GaLore: Memory-Efficient LLM Training by Gradient Low-Rank Projection) shows that the primary components could shift from one to another during the training. So, it’s better to examine the possibility of re-parametrizing the spectral adapters back into model weights, doing SVD again, and creating new spectral adapters to catch new primary components periodically during the training. By doing this we might come up with a better result.
>
> $\textbf{Answer to Questions.}$ Thanks the reviewer for raising up this question. In GaLore paper, the authors propose to re-SVD  after each $T$ iterations where $T$ is a free hyperparameter (page 5, section 4 in GaLore paper). The reason behind is that the authors suspect the spectral distribution of  "gradient" will shift while training. Overall, GaLore deals with subspace projection of gradient matrix instead of weight matrix itself.
>
> We note that it's more reasonable to assume there would be spectral shift for gradient matrices compared to weight matrices since gradients capture the fast descent directions which are more likely to change via training, while weight matrices are more invariant due to their strong impact on modeling capacity. For example, for vision models pretrained with human face pictures, when tuned with animal pictures, though there is likely to be some spectral space shift for weight matrices, we suspect it won't be large since human face pictures and animal pictures both obey real-world physics rules, from both structured shapes to color consistency.  While on the other hand, if we want to be more careful with spectral shift of weight matrices happened during training, we can employ similar method as GaLore and re-SVD after some number of iterations. Though the re-SVD procedure may bring some overhead.
>
>
> Please let us know whether our replies address all your concerns.  If so, could you please kindly consider increasing the score? If not, we are willing to address any other question in more details. Thanks!

---

> > ### Author Response · Authors · 2024-08-12
> >
> > Dear reviewer, since there are only two days left for author-reviewer discussion, we would like to confirm whether our responses have effectively addressed your concerns. We provided detailed responses to your concerns a few days ago, and we hope they have adequately addressed your issues. If you require further clarification on any of these points, please do not hesitate to contact us. We highly value the generous contribution of your time to review our paper. Thanks!

---

> > > ### Comment · Reviewer_N6x8 · 2024-08-13
> > > **Thanks for Response**
> > >
> > > Thanks for the authors' response. My main concerns have been addressed. Thus, I will keep my score.

---

### Official Review · Reviewer_Nthn · 2024-07-15

**Soundness:** 3
**Presentation:** 3
**Contribution:** 3
**Rating:** 6
**Confidence:** 4

**Summary:**

The paper presents a new low rank adapter for large models. The idea is to apply the adapter in the SVD decomposition of a weight matrix. Two methods are proposed. First, train parameters that get added to top r columns of U and V matrices. Second, train parameters that rotate top r columns of U and V matrices. The resulting methods are shown to perform well on LLMs and diffusion models.

**Strengths:**

I like the idea presented in the paper. It is intuitive and simple. A similar idea was presented recently in the PiSSA paper [1]. I think the experiments are sufficient and interesting. Moreover, compared to PiSSA that directly tunes the top r eigenvectors, additively tuning them has the benefit of providing a clear intuition when merging different adapters. I specially enjoyed reading the discussion on adapter merging as this is a topic that I have been thinking for a while now.


[1]: https://arxiv.org/abs/2404.02948

**Weaknesses:**

Evaluation is a big problem in this paper.

- LLM evaluations are few while diffusion evaluations seem to be mostly quantitative. I want to caveat this by saying that while few, the GSM-8k experiments with Mistral-7B are good enough to convince me that the method works. However, a comprehensive evaluation would have been more convincing and could shed light on cases where the method fails.
- I wish the authors had focused more on LLM evaluations. I may be biased as I work on LLMs.
- Figure 8 of supplementary (above like 586) is interesting and the authors should consider moving it to the main body of the paper.

**Questions:**

- Does training for more epochs change the results?
- Why did you only pick GSM8K?
- How do you accommodate that different methods may require different LRs to learn optimally?

**Limitations:**

The limitations section is missing. One limitation that I can think of is: is it possible for a model to overfit since we are always tuning the most important eigenvectors. What if the fine-tuning dataset is structured so that only a select non-top eigenvectors need to be updated. Datasets like GSM-8k restrict the model output to a narrow pattern that is significantly different from what the model outputs normally. Hence, updating the top eigenvectors makes sense. However, what if the fine-tuning dataset is supposed to modify only a few things that the model has learnt. For instance, there's a lora that switches the model output from Joe Biden to whoever is the next US president. I believe that updating the top eigenvectors is actually a problem here, and, as the authors note in lora merging section (lines 210-214), there  may need to be some eigenvector scheduling.

---

> ### Author Rebuttal · Authors · 2024-08-06
>
> Thanks for the careful review. Given the rebuttal character limit, we address $\textbf{most important questions}$ here, followed by some $\textbf{other comments later}$.
>
> $\textbf{Weaknesses 1.}$ LLM evaluations are few while diffusion evaluations seem to be mostly quantitative.
>
> $\textbf{Answer to Weaknesses 1.}$ Thanks for the suggestion.  To consolidate our work further, we have provided quantitative measurements for diffusion model results in our attached one-page pdf, and we have also appended these new results in a revision of our current paper.
>
> For LLM evaluations, we have both evaluated on GLUE benchmark with DeBERTa model (Table 1) and GSM8K benchmark with Mistral 7B (Table 2) and with Llama3 (Figure 1). Since GLUE benchmark is more about commonsense reasoning and GSM8K is for math tasks, we feel these cover common LLM reasoning tasks.  Moreover, the DeBERTa model we tried is of size 86M, Mistral model is of size 7B and Llama3 is of size 8B, thus we hope these cover both small and large models (given the resource constraint we have). We'll consider exploring more tasks in our revision.
>
> With respect to the quantitative evaluation for vision tasks, we follow the same computation metrics as used in paper [12] and paper [42], where the cosine similarity between clip vectors are taken for distance measurement.  Specifically, in ``Section A.1 in that pdf``, we provide quantitative results for our Figure 5. For text alignment score, we compute with the following prompts corresponding to each column:
>
>  1) "two dogs and a cat in front of Mount Fuji",
>
>  2) "two dogs and a cat in a galaxy",
>
>  3) "two dogs and a cat on a playground".
>
>  For image alignment score, since we are generating multi-character plots, we crop each generation vertically into three parts, with each cropped plot containing only a single animal. We then compute the alignment score of each  cropped component with corresponding reference images of the same animal.  For generation of column 1, FedAvg scores best but has only 0.0005 higher average than our method. For generation of column 2 and 3, our method achieves highest average and is around 0.01 better than other methods.
>
>
>  ``Section A.2 of the attached pdf`` contains quantitative results for our Figure 6. We include both the trend curve for alignment score vs. parameter budget (left panel), and also the exact score number (right panel, please expand to see the number). We also shade the region with trainable parameter budget only achievable by our method. Notably, it can be clearly observed that our method already generates sensible images with very few parameters ($\leq$ 20K) that can never be  obtained by other methods. For larger parameter budget, our method is able to generate images of quality  comparable to SOTA. This exemplifies the parameter efficiency of our method.
>
> $\textbf{Questions 1.}$ Does training for more epochs change the results?
>
> $\textbf{Answers to Question 1.}$  The plots in Figure 1 show training/testing results for different number of steps (though we only do single epoch training).  As what can be observed, the progress at the end of training is already pretty slow for all methods. Though theoretically we believe the global minimal training loss should be pretty close for all methods since $r=8$ is a small value and won't affect that much, the empirical progress can take long.
>
> $\textbf{Questions 2.}$ Why did you only pick GSM8K?
>
> $\textbf{Answer to Question 2.}$ As we have explained in our answer to Weaknesses 1, we evaluated on both GLUE and GSM8K with various model sizes. We hope this kind of  captures general reasoning capability. These are all metrics we have experimented with and we are not cherry-picking GSM8K. We prefer GSM8K to other specialized reasoning task since it's math problems and we are in STEM.
>
> $\textbf{Question 3.}$ How do you accommodate that different methods may require different LRs to learn optimally?
>
> $\textbf{Answer to Question 3.}$  We usually follow the lr setting in prior works, which we describe in each of the corresponding appendix sections. For example:
>
> 1. For Figure 1 results (GSM8K score), we follow  the parameter setting in QDoRA  blog https://www.answer.ai/posts/2024-04-26-fsdp-qdora-llama3.html, where they use the same lr for all LoRA, DoRA and QDoRA. We use the same setting for our spectral adapter and all baseline methods. We follow the setting there since we are training on the same dataset and the same model. This experiment is not covered in any of the primary report of our baseline methods.
>
> 2. For Table 1 results (GLUE score), we tune lr for our spectral adapter. For baseline methods, we follow hyperparameter setting for LoRA and AdaLoRA in their original reports for the same benchmark. We don't cite the score there since we are not tuning the exact same NN components/models. We use the same hyperparameter setting as LoRA for DoRA (since DoRA paper has no this benchmark evaluation) and we follow the setting used in BOFT, a variant of OFT, for OFT experiments. Since OFT primary report only includes vision tasks and BOFT instead compares to OFT on GLUE benchmark.
>
> 3. For vision tasks for adapter fusion, we use the default parameter setting for both our spectral adapter and all baseline methods as in original mix-of-show repo (https://github.com/TencentARC/Mix-of-Show) since our code is adapted from this repo.
>
> The reason why we generally don't tune each method individually is due to that we think granular lr value such as $2.2e-3$ used by AdaLoRA for GLUE (Table 8 in AdaLoRA's report) is hard to find and is thus impractical. We've tried hard to keep our comparisons fair for all methods. The lrs we have used for LLM tasks in the current paper ranging from $1e-3$ to $1e-5$, which cover common lrs used for fine-tuning LLMs.

---

> ### Author Response · Authors · 2024-08-06
>
> $\textbf{Strengths.}$   A similar idea was presented recently in the PiSSA paper.
>
> $\textbf{Comments on Strengths.}$ Thanks for mentioning PiSSA, which is a very recent and concurrent work to ours. This work explores a similar idea as our additive spectral adapter. We noticed it after completing our work. To distinguish our work from PiSSA and explain to other reviewers who may not be familiar with this line of work, we make several notes here. While PiSSA focuses only on LLM tasks and frames their method as a specific initialization method of classic LoRA model, we treat it quite differently and consider the fine-tuning procedure from a spectral decomposition angle. Here are some of the main components we consider which are not involved in PiSSA:
>
> 1) PiSSA does not consider rotational fine-tuning and does not maintain the orthogonality of SVD.
>
> 2) PiSSA does not explore adapter fusion, i.e., merging fine-tuned models.
>
> 3) PiSSA has no vision or diffusion model experiments.
>
> 4) PiSSA does not contain theoretic explanations.
>
> In contrast, we cover both additive and rotational SVD fine-tuning, adapter fusion by combining models fine-tuned via different spectral spaces, and we experiment with both LLM and generative diffusion models. Moreover, we have a theoretical analysis of adapter rank capacity (Lemma 3.1), weight subspace alignment (Section 3.2) and orthogonality via Cayley parameterization (Lemma 4.1). We believe there is still room for future research on NN weight spectral decomposition.
>
> Despite, our rotational spectral adapter has pushed parameter efficiency to certain extreme, in Figure 15 for example, we observe that our rotational adapter with $r=2$ is already performing quite well, which means each weight matrix is attached with only $2*2^2=8$ trainable parameters. As far as we know, this is the first PEFT model with trainable parameter budget scales with $r^2$ where $r$ is the rank, which is fully independent of weight dimension. Prior models such as VeRA and LiDB focus solely on reducing trainable parameter budget but their number of parameters still scales with original weight dimensions.
>
> $\textbf{Weaknesses 2.}$ I wish the authors had focused more on LLM evaluations. I may be biased as I work on LLMs.
>
> $\textbf{Answer to Weakness 2.}$  We'll consider involving more LLM tasks in our revision. We also feel LLM tasks seem more convincing since the quantitative evaluation metrics are more well-established.
>
> $\textbf{Weaknesses 3.}$ Figure 8 of supplementary (above like 586) is interesting and the authors should consider moving it to the main body of the paper.
>
> $\textbf{Answer to Weakness 3.}$ Thanks for the suggestion, we'll consider moving them to main text in our revision. We have also provided newly generated test scores for Figure 8 experiments, see the left and middle panel in  ``Section B of our attached one-page pdf`` for the results. It kind of shows that increasing the rank can close up the gaps between different methods.

---

> ### Author Response · Authors · 2024-08-06
>
> $\textbf{Limitations.}$  1. The limitations section is missing. One limitation that I can think of is: is it possible for a model to overfit since we are always tuning the most important eigenvectors. What if the fine-tuning dataset is structured so that only a select non-top eigenvectors need to be updated. Datasets like GSM-8k restrict the model output to a narrow pattern that is significantly different from what the model outputs normally. Hence, updating the top eigenvectors makes sense. However, what if the fine-tuning dataset is supposed to modify only a few things that the model has learnt. For instance, there's a lora that switches the model output from Joe Biden to whoever is the next US president. I believe that updating the top eigenvectors is actually a problem here, and, as the authors note in lora merging section (lines 210-214), there may need to be some eigenvector scheduling.
>
> $\textbf{Answer to Limitations.}$  Sorry about the missing of limitation section. We'll add a discussion about the limitations in our revision.  We feel the main potential limitation of the proposed method is still the overhead for computing SVD of weight matrices. However, for most modern networks (Stable Diffusion, Llama 3 and Mistral), this is tractable since the dimensions of layer weights are moderate. See Figure 7 which shows the SVD time/memory overhead is negligible. For certain other models with larger weights, our method may require more resources.
>
>
>
>
> With respect to the reviewer's concerns, we first would like to thank the reviewer for their careful considerations and detailed reading. We have actually thought about similar ideas around using non-principal spectral spaces before, we would like to explain some of our thoughts about this.
>
> First of all, if we have a fine-tuning task which  requires modification of non-top eigenvectors and the exact space needs fine-tuning is known a priori, we can apply spectral adapter to that particular subspace. In Figure 10 and 11 of Appendix F.5, we illustrate fine-tuning the bottom-$r$ eigenvectors for diffusion models, we have  added the results for fine-tuning starting at the $20$th column of $U$ and $V$ for Llama3 in the right panel of ``Section B of our attached one-page pdf``. We also  experiment with fine-tuning different singular vectors in our adapter fusion experiments (Section 4.2), where we assign different objects different spectral spaces (e.g., singular vector index 1 to 8 for object A and singular vector index 9 to 16 for object B). This prevents the overlap of the learned adapters in the weight space and improves the visual quality (Figures 5, 12 and 13).  We agree with the reviewer that some fine-tuning tasks may require modifying certain non-top eigenvectors, and knowing exactly which eigenvector to tune is usually pretty hard. A simple approach would be to run spectral adapter fine-tuning for different bands of singular vectors and evaluating them on the validation loss, then one can dynamically choose the best basis for fine-tuning. This can be done once at the beginning, or also after a fixed number of training iterations to accommodate for spectral shift that may happen.
>
>
> Second of all, while the training loss of fine-tuned models can be comparable for different adapters, their robustness levels can be different. Capacity wise, if there is a large change $\Delta W$ of low dimension required for weight $W$ while fine-tuning, then both LoRA component and our spectral adapter's component can be dedicated to model $\Delta W$.  Thus, these methods would have close training loss landscape.
> We feel that the difference between these adapter models would lie more in their robustness and stability in their optimization procedure. As shown in our analysis in Section 3.2, our spectral adapter can be more robust against optimization error in certain cases.
>
>
> Regarding the connection to eigenvector scheduling, we propose to schedule different tuning tasks over different spectral spaces thus to better utilize their orthogonality. We believe that modification happening within different orthogonal basis would affect each other less, this is also the underline logic of [42]. But say if we have an extreme case that task 1 is modifying exactly the $5$-th column of $U$ and $V$, and task 2 is modifying exactly the $10$-th column of $U$ and $V$. Then in this case, we agree that it would be ideal if one could recognize a priori which indices need to be tuned, though this is a nontrivial job. Our hope here is that, with rank $r=1$ for example, tuning the top component would capture a bit about task 1 (the top part may change from $U_1S_1V_1^T$ to some combination of $U_1S_1V_1^T$ and $U_5S_5V_5^T$), and tuning the second top component would capture a bit about task 2. These differences would happen more independently with respect to each eigen basis. Thus the integrity of each task is better preserved.

---

> ### Author Response · Authors · 2024-08-06
>
> Please let us know whether our replies address all your concerns. If so, could you please kindly consider increasing the score? If not, we are willing to address any other question in more details. Thanks!

---

> ### Author Response · Authors · 2024-08-13
>
> Dear reviewer, given that this is the last day of author-reviewer discussion period, we would like to for the last time check whether our above responses have addressed your prior concerns satisfactorily. Hope that given our (newly) incorporated quantitative scores for our diffusion model experiments, which demonstrate improvement of our method, and different models/benchmarks for LLM tasks we've already considered, your prior concerns with respect to evaluation problems have been alleviated.
>
> If our responses effectively tackled all your questions,  could you please kindly consider raising your score? If there is any point requires further classification, please let us know and we'll be here to address any existing questions until the final second of the discussion period. Thanks again for spending time reviewing our work! We highly value the generous contribution of your time to review our paper.

---

### Author Rebuttal · Authors · 2024-08-06

Since there are both similar questions that different reviewers have asked about and some points we want to make clear to all reviewers, we summarize several important points here. We also reply individually to each reviewer with respect to their own concerns in more details. We have highlighted the newly added content in the one-page pdf attached below, and all Sections/Figures mentioned in our reply which are not highlighted can be found in our original paper.

$\textbf{Quantitative evaluations for diffusion model results:}$ as suggested by both reviewer 7N9o and reviewer Nthn, we have added quantitative evaluations for diffusion model experiments in ``Section A of our attached one-page pdf``. We follow evaluation metrics in paper [12] and [42] to compute cosine similarity between clip vectors. From the results it can be observed that out method hits best overall average (Section A.1) and generates good results with small parameter budget (Section A.2, we shade the region of parameter budget only achievable by our method).

$\textbf{Tuning non-top spectral space:}$ as pointed out by both reviewer Nthn and reviewer N6x8, there might be scenarios that only some non-top spectral space needs  fine-tuning, and there might also be spectral distribution shift during fine-tuning. We note that first of all, if one knows a priori which part of the spectral space needs fine-tuning, one can directly apply the proposed spectral adapter method to it. We provide tuning bottom$-r$ singular vector results in Figure 9, Figure 10, and Figure 11. We have also provided additional results for fine-tuning Llama3 8B starting at $20$th column of $U,V$ matrices in ``Section B (right most panel) of our attached one-page pdf``, from which it can be observed that tuning top spectral component behaves better. Moreover, in Section 4.2, we study adapter fusion by distributing each tuning task to different spectral spaces. While knowing which index of the singular vector matrix needs fine-tuning a priori  is usually hard, some more advanced scheduling strategy can be adopted. For example,  one can run spectral adapter fine-tuning for different bands of singular vectors and evaluate them on the validation loss, then one can dynamically choose the best basis for fine-tuning. This can be done once at the beginning, or also after a fixed number of training iterations. To deal with spectral shift while training, re-SVD after some fixed number of iterations can tackle this.

---

> ### Author Response · Authors · 2024-08-12
>
> $\textbf{Correction to image scores in Section A.1 of attached pdf.}$ We note that in Section A.1, we get all scores by adapting from the codebase of textual inversion repo (https://github.com/rinongal/textual_inversion/tree/main), due to rebuttal time limit,  the only change we made is that we cropped each generated plot vertically into three columns, matched with each reference animal, and computed the average scores.
>  The result indicates that our spectral adapter gains uniform improvement for text alignment scores over all three scenarios. However, for image scores, our method is inferior in some cases, though the average score of our method is still pretty good.  The worse image alignment score in some cases is a bit  counterintuitive, for column 2 (the galaxy scenario) for example, our generation is visually better than Gradient Fusion and FedAvg in capturing animal shapes, but our image score is worse.
>
> To further investigate this issue, we do a finer crop and recompute the score. We find the prior inferiority of our method's image score is due to that the animals we generate are sometimes vertically smaller, and the background is sometimes more colorful/distractive, e.g., see our generation and FedAvg's generation for the galaxy scenario for comparison. This negatively influenced the score computed and made the score inaccurate. After we finer-crop each column further and zoom in to focus more on the animals, we get the following revised image scores (text scores remain the same since we always compute with respect to the full plot). We'll release our evaluation code as well. Note all methods have higher image scores this time:
> $$
> \qquad\qquad\qquad\qquad\qquad\qquad\quad\text{ Gradient Fusion }   \qquad\qquad  \text{ FedAvg}   \qquad \qquad  \quad      \text{Orthogonal Adaptation}\qquad   \text{Spectral Adapter (Ours) }
> $$
> $$
> \text{ Mount Fuji Scenario} \qquad \qquad 0.8906 (Avg. 0.5945) \qquad  0.8716  (Avg. 0.5942) \qquad\quad  0.8423  (Avg. 0.5913) \qquad  0.8540  ~(Avg. 0.6007\uparrow)
> $$
> $$
> \text{ Galaxy Scenario} \qquad \qquad\qquad 0.8301 (Avg. 0.5524) \qquad  0.8521  (Avg.0.553 )\qquad\quad~~  0.8120  (Avg.0.5614 )\qquad  0.8560\uparrow  (Avg.5859 \uparrow)
> $$
> $$
> \text{Playground Scenario} \qquad \quad\quad ~0.8711 (Avg.5732 ) \qquad\quad  0.8765  (Avg. 0.5712)\qquad\quad  0.8530  (Avg. 0.5712)\quad~~~  0.8892\uparrow  (Avg.0.5901\uparrow)
> $$
> As we can see, our method improves both text and image scores uniformly for the last two scenarios, and achieves the best average scores in all three scenarios.

---

### Decision · Program_Chairs · 2024-09-25

**Decision:**

Accept (poster)

**Comment:**

The paper  introduces an interesting and useful idea, to use SVD in providing a Low-rank approximation to expensive network operations. The paper provides very good theoretical analysis and explanation of the approach.

Most of the primary issues raised by the reviewers were handled in the rebuttal and subsequent discussion, and two of the reviewers raised their scores as a result. A minor weakness of the method is the added computational expense of performing the SVD.  The rebuttal makes the case that the overhead is minimal compared to standard LORA, but there still is an expense. In figure 7 of the paper the increase in size and runtime compared to LORA appears to be about 10% (not negligible) The caching of the SVD results might not be possible, due to scaling, depending on the particular application and particulars of the hardware being used.

The strengths of the paper lie in the simple, yet powerful, spectral network adaptor approach, which produces enhanced performance compared to competing methods on diffusion model and LLM fine-tuning problems. The proposed spectral approach is a new take on the adaptor style of parameter efficient fine tuning methods. Theoretical results are provided that show the spectral adaptor method to have higher rank capacity than Lora adaptors.
The adaptor fusion approach is novel and interesting. The proposed method operates on orthogonal singular vectors and thus results in an elegant solution to multi-adapter fusion problems by distributing different concept tunings along different columns of singular vector matrices.